# Niacin restriction with NAMPT-inhibition is synthetic lethal to neuroendocrine carcinoma

Miyuki Nomura[1], Mai Ohuchi[1], Yoshimi Sakamoto[1], Kei Kudo[1,2,3], Keisuke Yaku[4], Tomoyoshi Soga [5], Yuki Sugiura[6], Mami Morita[1], Kayoko Hayashi[1], Shuko Miyahara [1,2,3], Taku Sato[1], Yoji Yamashita[1], Shigemi Ito[1], Naohiko Kikuchi[1], Ikuro Sato[7], Rintaro Saito[5], Nobuo Yaegashi[3], Tatsuro Fukuhara [8], Hidekazu Yamada[1], Hiroshi Shima[1], Keiichi I. Nakayama [9,10], Atsushi Hirao[11], Kenta Kawasaki[12], Yoichi Arai[13], Shusuke Akamatsu[14,15], Sei-ichi Tanuma[16,17], Toshiro Sato [12,18], Takashi Nakagawa [4] & Nobuhiro Tanuma [1,2] ✉

Nicotinamide phosphoribosyltransferase (NAMPT) plays a major role in NAD biosynthesis in many cancers and is an attractive potential cancer target. However, factors dictating therapeutic efficacy of NAMPT inhibitors (NAMPTi) are unclear. We report that neuroendocrine phenotypes predict lung and prostate carcinoma vulnerability to NAMPTi, and that NAMPTi therapy against those cancers is enhanced by dietary modification. Neuroendocrine differentiation of tumor cells is associated with down-regulation of genes relevant to quinolinate phosphoribosyltransferase-dependent de novo NAD synthesis, promoting NAMPTi susceptibility in vitro. We also report that circulating nicotinic acid riboside (NAR), a non-canonical niacin absent in culture media, antagonizes NAMPTi efficacy as it fuels NAMPT-independent but nicotinamide riboside kinase 1-dependent NAD synthesis in tumors. In mouse transplantation models, depleting blood NAR by nutritional or genetic manipulations is synthetic lethal to tumors when combined with NAMPTi. Our findings provide a rationale for simultaneous targeting of NAR metabolism and NAMPT therapeutically in neuroendocrine carcinoma.

Although many metabolic activities are conserved among cancers, several metabolic characteristics are specific to particular cancer types[1–4]. The latter are often associated with either particular genetic alterations or the tissue of origin. Some cancer type-specific metabolic activities are marked by extremely high demand for a particular metabolic pathway and/or nutrient, which if targeted would present a metabolic vulnerability. For example, cells from relapsed specimens or stem-cell fractions of acute myeloid leukemia (AML) reportedly show high dependence on NAMPT, a rate limiting enzyme in the NAD salvage pathway, relative to either de novo AML or normal hematopoietic stem cells[5,6].

NAD (NAD+ and NADH) is an essential cofactor that drives many metabolic reactions. NAD is continuously degraded by NAD consumers such as Sirtuins and PARPs and also consumed as a substrate for NADPH synthesis. Thus, constant biosynthesis is essential to maintain an adequate cellular NAD pool. In vertebrates, NAD can be built from Trp or niacin-class vitamins in cells that can metabolize those factors. Changes in NAD pool size have consequences for health and are seen in various diseases, including cancer[7]. A recent report revealed that excessive NAD pool formation triggers immortalization of tumor-initiating cells from Drosophila brain tumors[8]. However, our

---

understanding of the dependence of various cell types on NAD biosynthesis or how precursor niacin regulates NAD pool size is limited.

We previously reported that small cell lung cancer (SCLC) is unique in expressing PKM1, a hyper-active isoform of the glycolytic enzyme PKM, and that PKM1 is required for SCLC cell survival and proliferation[4]. PKM1 promotes glucose metabolism more efficiently than does the PKM2 isoform; yet, it is unclear how PKM1-directed active glucose metabolism supports SCLC. Here we analyzed metabolic advantages conferred by PKM1 expression in this cancer context and found that PKM1 promotes NAD biosynthesis. These findings prompted us to evaluate NAD metabolism in SCLC. We found that SCLC and other neuroendocrine carcinomas (NECs) were vulnerable to NAMPT inhibition. We also reveal that in mice, dietary niacin counteracts the efficacy of targeting NAMPT in those cancers.

## Results

### PKM1, relative to PKM2, activates NAD synthesis in mouse cells

We first examined metabolic difference(s) using cultured lung epithelial (LE) cells and MEFs derived from mice genetically engineered to express either PKM1 or PKM2 ($Pkm^{M1/M1}$ or $Pkm^{M2/M2}$)[4]. In both cases, cells were immortalized and transformed with oncogenic KRAS (Fig. 1a). In LE cells, steady-state NAD (NAD+ and NADH) levels and protein PARylation, which requires NAD as a substrate, were higher in cells expressing only PKM1 than in PKM2-expressing cells (Fig. 1b). NAD levels in MEFs became transiently higher in PKM1 relative to PKM2 cells after a medium change to replenish glucose (Fig. 1c).

NAD can be synthesized from nutrients including Trp and niacin-class vitamins, although most general culture media contain only Trp and nicotinamide (Nam) (Fig. 1d). Tracer experiments using $^{15}$N-Nam showed that >90% of NAD in $Pkm^{M1/M1}$ MEFs was synthesized from Nam, and that Nam-to-NAD flux was significantly higher in $Pkm^{M1/M1}$ than in $Pkm^{M2/M2}$ cells (Fig. 1e, f). We were unable to detect substantial NAD labeling when $^{13}$C-Trp was used as a tracer. NAD synthesis from Nam (known as NAD salvage) requires Nam, PRPP and ATP as substrates. Both total and glucose-derived PRPP and ATP levels were higher in $Pkm^{M1/M1}$ than $Pkm^{M2/M2}$ cells after a medium change (Fig. 1g, h). Collectively, these results show that relative to PKM2, PKM1 more efficiently activates the NAD salvage pathway, likely by increasing glucose flux to two essential substrates, PRPP and ATP (Fig. 1i). Since unlike PKM2, PKM1 activates both glucose anabolism and catabolism[4], the latter may facilitate ATP regeneration from ADP.

### SCLC shows significant vulnerability to inhibition of NAD salvage

To determine the role of NAD salvage in SCLC, we first asked which synthetic pathway(s) is essential to maintain NAD levels in SCLC. Knockdown of various metabolic enzymes in SCLC cells suggested that only the salvage pathway, in which NAMPT is the rate-limiting enzyme, is essential to maintain NAD levels in normal culture conditions (Fig. 2a, b and Supplementary Fig. 1a–c).

We next compared responses of SCLC and non-small cell lung cancer (NSCLC) lines to the NAMPT inhibitor (NAMPTi) FK866. FK866 treatment of SCLC cells resulted in robust cell death, which was rescued by including NMN, a metabolite downstream of NAMPT, in culture media (Fig. 2c and Supplementary Fig. 1d, e). These observations exclude the possibility of off-target effects. In contrast, most NSCLC lines were resistant to FK866-induced cell death (Fig. 2c and Supplementary Fig. 1e). We obtained similar results following treatment of lung cancer lines with two other NAMPTis, GNE-617 or TLM-118 (Fig. 2c). Moreover, SCLC line sensitivity to FK866 was comparable in 4 AML lines that reportedly show high NAMPT dependence (Supplementary Fig. 1e, f)[5,6]. NAMPT knockdown by siRNA also suppressed proliferation of all 3 SCLC lines tested (Supplementary Fig. 1g). Analysis of the DepMap dataset[9–11] revealed that *NAMPT*-KO effects in SCLC were more robust than in pan-cancer or NSCLC groups (Fig. 2d) but

similar to the AML group (Supplementary Fig. 1h). Collectively, we conclude that SCLC cell survival and proliferation is highly dependent on the NAD salvage pathway.

Cultured SCLC cells treated with FK866 showed NAD depletion within 48 h (Fig. 2e), and metabolome analysis revealed NAD to be the most significantly decreased metabolite following FK866 treatment (Fig. 2f). FK866 treatment of SCLC cells also promoted (1) decreases in levels of high-energy nucleotides (ATP, GTP, UTP and UTP), suggestive of energy deficiency, and (2) accumulation of F1,6BP, DHAP and IMP, suggesting defects in glycolysis and purine synthesis. Tracer experiments using $^{13}$C-glucose showed that FK866 treatment blocked glucose metabolism at steps catalyzed by GAPDH, IMPDH and ADSS (Fig. 2g and Supplementary Fig. 2a). Both GAPDH and IMPDH are NAD-dependent and their inactivation is consistent with NAD depletion. We hypothesized that GAPDH inactivation would promote energy deficiency, as GAPDH inactivation converts glycolysis from an energy-producing to an energy-consuming pathway (Supplementary Fig. 2b). To test this possibility, we treated SCLC cells with the small molecule GAPDH inhibitor koningic acid (KA) and observed a glucose-dependent decrease in ATP levels, a finding consistent with a previous report that KA treatment causes glucose-dependent ATP decreases in susceptible cells[12] (Fig. 2h and Supplementary Fig. 2c). These results suggest a model in which NAMPTi treatment induces NAD depletion in SCLC cells, which inactivates GAPDH and promotes energy deficiency and subsequent cell death (Fig. 2i). We also note that this activity would create a feedforward loop in which ATP decreases lead to further NAD decline, as ATP is an essential substrate in NAD salvage.

### Absence of QPRT-dependent de novo NAD synthesis promotes NAMPTi-susceptibility in SCLC

We next assessed metabolic difference(s) in cancers sensitive and insensitive to NAMPTi by comparing NSCLC and SCLC lines. FK866 treatment strongly suppressed NAD levels in SCLC lines but only moderately in NSCLC lines (Fig. 3a). NAD levels seen after FK866 treatment of NSCLC cells differed from line to line (ranging from 7 to 48%). Nevertheless, the energy status of FK866-treated SCLC and NSCLC lines differed markedly, with treatment severely depleting ATP in SCLC but not NSCLC lines (Fig. 3b). In SCLC cells, FK866 treatment caused slight decreases in adenylate and significant changes in guanylate energy-charge, and comparable effects were generally not seen in NSCLC lines (Fig. 3c). These results suggest that energy status is a good predictor of NAMPTi susceptibility. Unlike SCLC, most KA-treated NSCLC lines maintained normal ATP levels (Supplementary Fig. 2d). Collectively, we conclude that most NSCLC lines show either (i) an NAMPT-independent ability to maintain the NAD pool or (ii) a GAPDH-independent ability to maintain energy status, or both. Conversely, SCLCs are deficient in both and hence are highly vulnerable to NAMPT inhibition.

To assess the basis of differential NAMPTi susceptibility, we investigated expression of mRNAs relevant to both NAD de novo synthesis and metabolism of Trp to NAD in SCLC and NSCLC lines (Fig. 3d, e). Analysis of 50 SCLC and 135 NSCLC lines in the DepMap dataset revealed significantly lower transcript levels of 3 genes in SCLC compared to NSCLC lines (Fig. 3f), with *KYNU* showing the most significant decrease. We then selected two NSCLC lines (H1975 and LC-KJ) that showed high KYNU transcript levels and used siRNA to knockdown either QPRT, which catalyzes the final step of the de novo pathway, or KYNU itself. *KYNU* knockdown gave mixed results, while in both lines *QPRT* knockdown decreased NAD levels before and after FK866 treatment (Fig. 3g, h and Supplementary Fig. 2e, f). These results are in contrast to SCLC, in which *QPRT* knockdown had almost no effect (Fig. 2b). Furthermore, ectopic QPRT expression in 87-5 and NCI-H209 SCLC cells conferred FK866-resistance based on observations of NAD levels and cell proliferation (Fig. 3i–k). These results

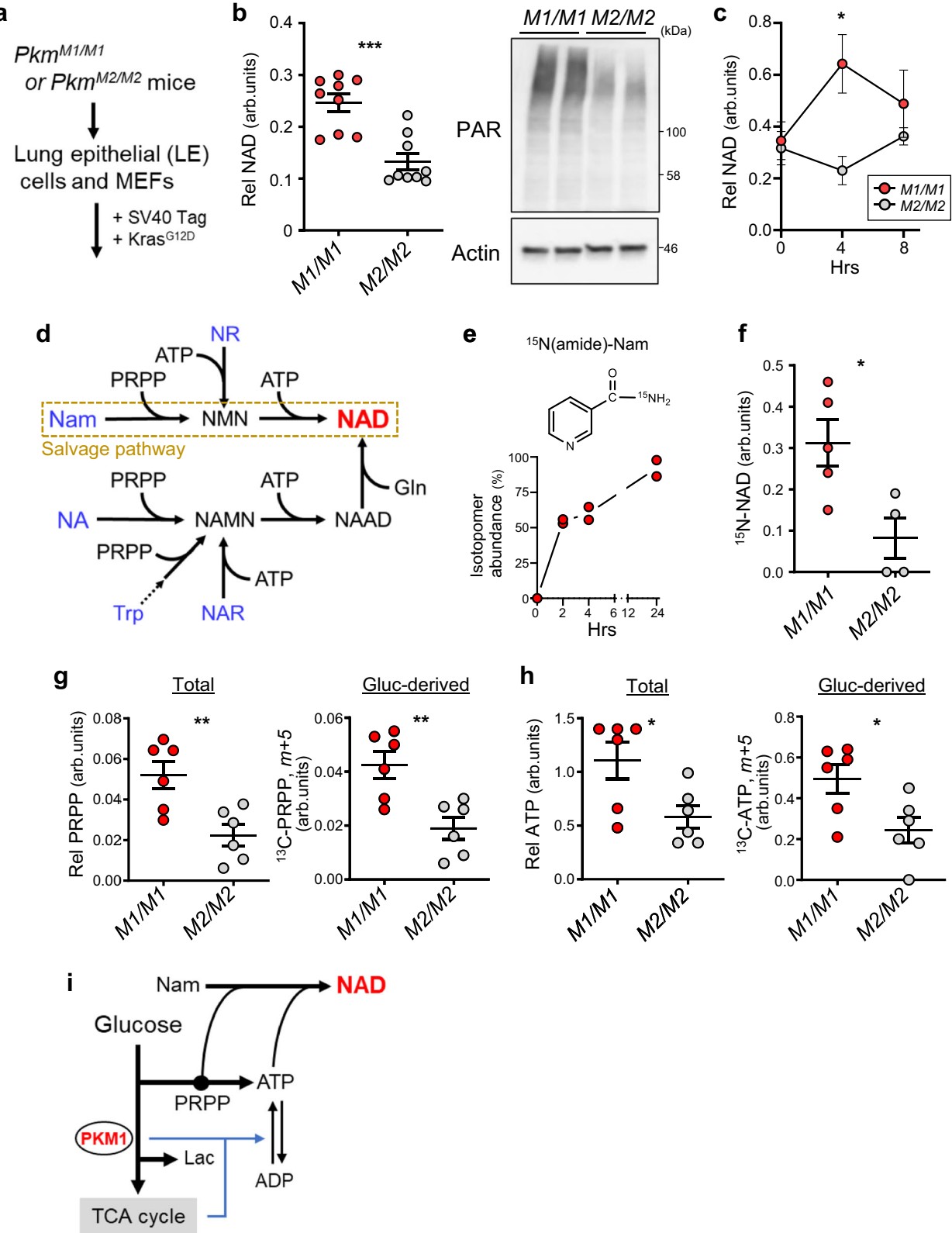

indicate that de novo NAD synthesis, if activated, compensates for NAMPT inhibition in SCLC, and suggest that this activity partially accounts for NAMPTi-resistance in some NSCLC cells.

**High NAMPT-dependence is associated with NE phenotypes**
We next asked whether NAMPT dependence is associated with neuroendocrine (NE) differentiation, since SCLC is typically a NE

carcinoma (NEC)[13]. NE differentiation of the engineered organoid TR-6TF[14] established from human intestine was accompanied by increases in the PKM1/PKM2 mRNA ratio, decreases in transcript levels of two genes (including KYNU) relevant to NAD de novo synthesis, and NAMPTi susceptibility, based on analysis of NAD and ATP levels (Fig. 4a–f and Supplementary Fig. 3). These results show that NAMPT dependence is directly linked to NE differentiation.

**Fig. 1 | PKM1 activates NAD synthesis in mouse cells more robustly than PKM2. a** Isolation and transformation of *Pkm*[M1/M1] or *Pkm*[M2/M2] LE cells and MEFs. **b** Steady state levels of cellular NAD (left) and poly-ADP-ribosylated protein (PAR) (right) in *Pkm*[M1/M1] and *Pkm*[M2/M2] LE-Kras cells. *n* = 3 biological replicates from 3 LE lines for each genotype. *P* = 0.0002. The blot shown is representative of three independent experiments. **c** NAD levels in *Pkm*[M1/M1] and *Pkm*[M2/M2] MEF-Kras cells before and after medium change (4 or 8 h). *n* = 6 (0 h), 4 (4 h, *M2/M2*) or 5 (others) biological replicates from two MEF lines for each genotype. *P* = 0.0197 (4 h). **d** Diagram of NAD synthetic pathways in mammals. Nam nicotinamide, NA nicotinic acid, NAR nicotinic acid riboside, NR nicotinamide riboside, NMN nicotinamide mononucleotide, NAMN nicotinic acid mononucleotide, NAAD nicotinic acid adenine dinucleotide. **e** Enrichment of labeled NAD (labeled vs total NAD) in *Pkm*[M1/M] MEF-Kras cells cultured in the presence of [$^{15}$N (amide)]-Nam for indicated times. Shown are data from two independent MEF lines. **f** NAD synthesis from Nam in *Pkm*[M1/M1] and *Pkm*[M2/M2] MEF-Kras cells. Cellular $^{15}$N-NAD (NAD, m+1) levels were measured 4 hours after $^{15}$N-Nam loading. *n* = 5 (M1/M1) or 4 (M2/M2) biological replicates from two MEF lines per genotype. *P* = 0.0202. **g** Levels of total (left) and glucose-derived (right) PRPP (m+5) in *Pkm*[M1/M1] and *Pkm*[M2/M2] MEF-Kras cells loaded 4 h in medium containing [U-$^{13}$C]-glucose. Shown are the average of three independent MEF lines per genotype with duplicates. *P* = 0.0062 (Total); *P* = 0.0047 (Gluc-derived). **h** Levels of total (left) and glucose-derived (right) ATP, determined as in (**g**). Shown is the average of three independent MEF lines per genotype, with duplicates. *P* = 0.0269 (Total); *P* = 0.0248 (Gluc-derived). **i** Potential crosstalk between glucose-metabolism and NAD synthesis. Data are presented as mean plus the range (**e**) or SEM (**b**, **c**, **f**, **g**, **h**). *$P$ < 0.05, **$P$ < 0.01, ***$P$ < 0.001 as determined by two-tailed *t* test (**b**, **c**, **f**–**h**). Source data are provided as a Source Data file.

SCLC and small-cell prostate cancers (SCPCs) show significant similarity in terms of NE differentiation[15]. Thus, we next asked whether NAMPTi susceptibility varies according to prostate cancer (PCa) sub-type. Recently, NE differentiation of prostate adenocarcinoma (AdPCa) (known as trans-differentiation (or lineage plasticity)) was reported as a mechanism underlying AdPCa resistance to endocrine therapy and emergence of castration-resistant PCa (CRPC)[16–19]. After classifying PCa lines as either PCa with NE differentiation (NE-PCa) or "others", we found that NE-PCa lines were more susceptible to FK866 than were other PCas, as seen in lung cancer (Fig. 4g–j). SCLC and NE-PCa lines also showed significantly higher susceptibility to NAMPTi compared to several human non-transformed cell lines (Fig. 4k).

### Restriction of dietary niacin enhances NAMPT-targeting therapy in mice

Studies in preclinical models indicate that combining specific cancer therapies with dietary interventions impacts therapeutic efficacy[20–24]. Two major dietary precursors of NAD are Trp and niacin-class vitamins: upon absorption, Trp in liver is converted to Nam, a substrate for NAD salvage, and then released into the circulation[25], although how dietary niacin is metabolized systemically is unclear. We used mouse models to determine whether NAMPT inhibition could serve as anti-cancer therapy and established two synthetic diets depleted of either niacin (NFD, or a niacin-free diet) or tryptophan (WFD). We then fed mice bearing Lu-139 SCLC tumors either a normal diet, the NFD or the WFD and administered the NAMPTi GNE-617 to all three groups (Supplementary Fig. 4a, b). Administration of GNE-617 to WFD mice promoted rapid loss of body weight, prompting us to discontinue the WFD at day 3. Combining GNE-617 with a NFD resulted in a small and transient loss of body weight but, importantly, had the most powerful anti-tumor activity relative to all other groups (Fig. 5a and Supplementary Fig. 4a, b). Combining GNE-617 with a NFD synergistically decreased tumor NAD levels (Fig. 5b and Supplementary Fig. 4c). We also found that combining a WFD with GNE-617 treatment synergized to decrease NAD levels and tumor growth (Fig. 5a and Supplementary Fig. 4d), despite the observed high toxicity. Combining GNE-617 with a NFD significantly suppressed growth of SCLC (Lu-139), SCPC (NCI-H660) and NE-CRPC (KUCaP13) tumors in xenograft models compared to either control or GNE-617/normal diet groups (Fig. 5c). We judged NFD effects in combined therapy to be rapid, as use of a simultaneous protocol (in which diet was changed immediately after drug administration) significantly reduced levels of tumor NAD relative to GNE treatment alone (Supplementary Fig. 4e). Moreover, effects were comparable when we employed either the simultaneous protocol or a protocol in which mice were fed the NFD for 3 days before GNE administration (Supplementary Fig. 4e).

We then assessed the tumor metabolome in the presence of NFD, GNE-617 or their combination (Supplementary Fig. 5a). Although GNE-617 alone decreased NAD in mice fed a normal diet, its combination with a NFD significantly increased F1,6BP and DHAP and promoted a severe energy deficiency (Supplementary Fig. 5b–e). Time-course metabolome analysis showed that the effect of NFD alone was minimal but that the metabolome changed as combination therapy continued over time (Supplementary Fig. 5f, g). Specifically, after 2 days of GNE-617 treatment, we observed almost complete NAD depletion plus accumulation of F1,6BP and DHAP, and by the final day of treatment mice exhibited an energy deficiency (Supplementary Fig. 5h–k). Thus, metabolic changes seen in tumors receiving combined therapy resembled those seen in NAMPTi-treated cells.

Feeding with a NFD enhanced anti-tumor effects not only of GNE-617 but of TLM-118[26], a different NAMPTi (Fig. 5d). GNE-617/NFD combined therapy was highly effective in a variety of SCLC and SCPC xenograft models, although NSCLC xenograft tumors tested were almost resistant to combined therapy (Supplementary Fig. 6a–c).

### Niacin restriction lowers blood NAR and restricts NAMPT-independent NAD synthesis in tumor cells

We next investigated mechanisms underlying synthetic lethality seen after combined NFD/ NAMPTi treatment. Previous studies had suggested that circulating NA and/or NR antagonize NAMPTi effects on tumors[27,28]. We observed that such effects of NR were less potent than those of NA and NAR in NAMPTi-treated SCLC and SCPC cells (Supplementary Fig. 7a, b). Furthermore, phenotypes seen in cultured KUCaP13 cells were not rescued by NA treatment, despite the strong NFD/NAMPTi synergy seen in corresponding xenograft models (Figs. 5c and 6a and Supplementary Fig. 7c). Given that these results question the importance of NA and NR, we assessed levels of 10 molecules (all suggested to have niacin activity[29] (see "Methods")) in blood of mice fed either a normal diet or NFD and detected four types of niacin molecules (Nam, NA, NR and NAR; Fig. 6b). Relative to a normal diet, a NFD specifically lowered NAR levels (Fig. 6c and Supplementary Fig. 7e), suggesting that NFD/NAMPTi synergy results from selective depletion of circulating NAR. On the other hand, serum NAR and Nam levels decreased in mice fed a WFD (Supplementary Fig. 7f).

NAR supplementation of cell culture medium rescued all cell lines tested (including KUCaP13) from FK866-mediated cytotoxicity (Fig. 6a and Supplementary Fig. 7c, d). Moreover, we knocked out *NAPRT* in SCLC lines to obtain lines that can utilize NAR but not NA for NAD synthesis (Fig. 6d, e and Supplementary Fig. 8a–d). Tumors derived from those lines showed no signs of enhanced GNE-617 susceptibility relative to parental cells. Rather, NFD strongly synergized with NAMPTi, even against *NAPRT*-KO tumors, excluding the possibility that NFD effects are related to a tumor's NA-metabolizing ability (Fig. 6f and Supplementary Fig. 8e). On the other hand, synergistic effects of NFD and GNE-617 on tumor NAD levels in mice were completely blocked by NAR administration (Fig. 6g and Supplementary Fig. 8f). Collectively, these results suggest that NFD effects are primarily attributable to lowering of blood NAR levels. NMRK1 loss via either gene knock-out or knockdown decreased cellular NAD levels in FK866-treated, NAR-

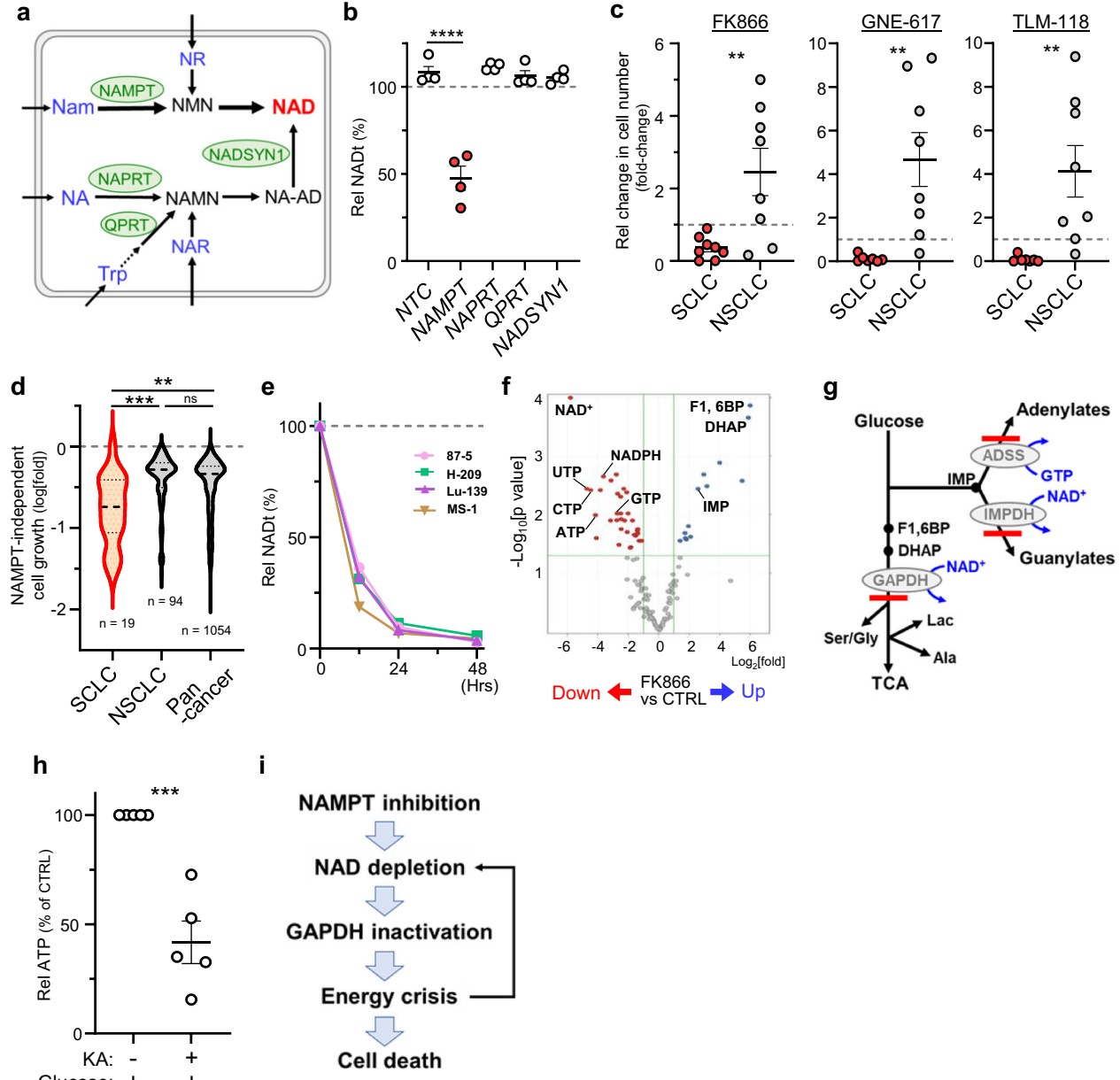

**Fig. 2 | Significant vulnerability of SCLC to inhibition of NAD salvage. a** Key metabolic enzymes involved in NAD biosynthesis (shown in green) and potential nutrient sources (blue) taken up by cells. Note that Nam and Trp are the only NAD precursors contained in normal culture medium. See also Fig. 1, which describes substrates required for each reaction. **b** Effects of siRNA-mediated knockdown of enzymes shown in (**a**) on NAD levels in 4 SCLC lines. Shown are values relative to those seen in mock-transfected cells, which were defined as 100%. Circles represent individual lines. $P < 0.0001$ (NTC vs NAMPT). **c** Comparison of relative numbers of cells (relative to the number on day 0) in SCLC and NSCLC lines cultured 4 days with indicated NAMPT-inhibitors (FK866, GNE-617 or TLM-118), each at 20 nM. Each symbol represents an individual cell line. Shown are results of 8 (for FK866) or 7 (GNE-617 and TLM-118) SCLC lines and 8 NSCLC lines. $P = 0.0069$ (FK866); $P = 0.0045$ (GNE-617); $P = 0.0073$ (TLM-118). **d** Analysis of the DepMap dataset showing effects of *NAMPT* knock-out on proliferation of cell lines in the CCLE collection. Results were compared in pan-cancer, NSCLC and SCLC categories. $P = 0.0001$

(SCLC vs NSCLC); $P = 0.0017$ (SCLC vs Pan-cancer); $P = 0.0652$ (NSCLC vs Pan-cancer). **e** Time course analysis of NAD levels in 4 NAMPTi-treated SCLC lines (87-5, H209, Lu-139, and MS-1). Shown is data representative of >2 independent experiments. **f** Volcano plot of the metabolome of 4 SCLC lines in (**e**) treated 48 h with or without NAPMTi. Significantly increased or decreased metabolites compared to untreated controls are shown in blue and red, respectively. *p* values were based on a two-tailed *t* test. **g** Summary of [13]C-glucose tracer experiments in SCLC cells. FK866 treatment blocked glucose metabolism at steps catalyzed by GAPDH, IMPDH and ADSS (thick red lines). Lac., lactate. TCA TCA cycle. **h** ATP levels in 5 SCLC lines treated 6 h with Koningic acid (KA) (a GAPDH inhibitor) or control vehicle. $P = 0.0003$. **i** Proposed model of NAMPT inhibition-induced death of SCLC cells. Data are presented as the mean of measurement duplicates (**e**) or the mean of biological replicates plus the SEM (**b**, **c**, **h**). **$P < 0.01$, ***$P < 0.001$, ****$P < 0.0001$ as determined by one-way ANOVA with a post hoc test (**b**, **d**) or by two-tailed *t* test (**c**, **h**). Source data are provided as a Source Data file.

supplemented tumor cells (Fig. 6h, i and Supplementary Fig. 8g), as did *NADSYN1* knock-out (Fig. 6j, k). These findings are consistent with previous studies reporting that NMRK1 (or NMRK2 in some cell types) is required for metabolism of NAR to NAMN, which fuels NAMPT-independent NAD synthesis[30,31] (Fig. 6l).

## Blood NAR is derived from dietary NA, NAPRT-dependently
The normal rodent diet used here contained Nam, NA and NR but not NAR (Fig. 7a). We fed mice various NFDs supplemented with specific niacin molecules and found NA necessary and sufficient for re-establishing normal blood NAR levels and negating the effects of a NFD

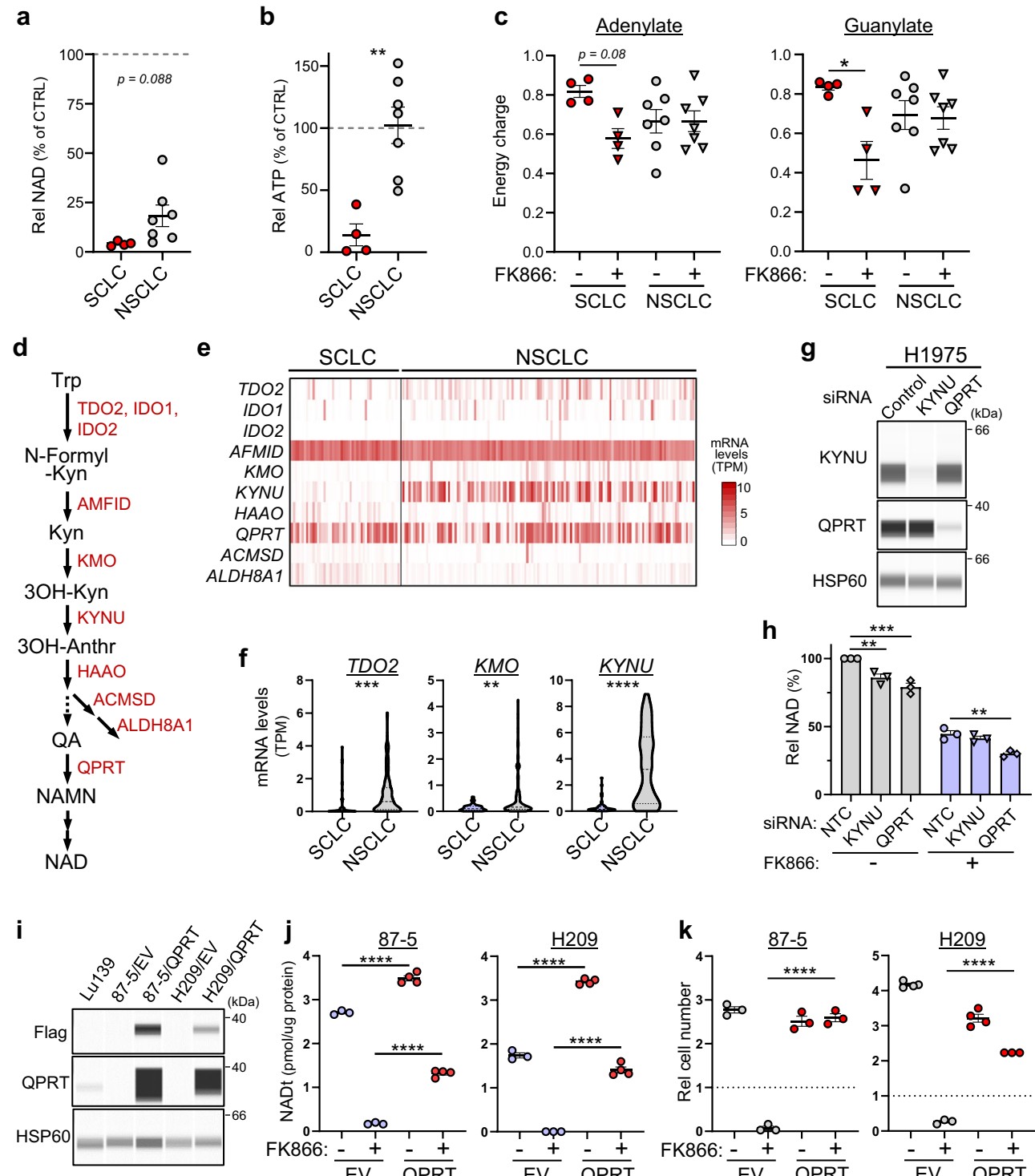

(Fig. 7b and Supplementary Fig. 9a–c). NA administration by gavage to NFD-fed mice rapidly increased blood NA and NAR levels (Fig. 7c). When we repeated these experiments using deuterium-labeled NA, blood NAR was extensively labeled and NAR forms *m+4* and *m+3* accounted for ~99% of total NAR (Fig. 7d). These results demonstrate that circulating NAR is synthesized from dietary NA.

Given rapid NAR elevation in blood after NA administration and the labeling pattern seen in tracer experiments, we hypothesized that dietary NA is converted to NAR via NA -> NAMN -> NAR, and that the first step is catalyzed by NAPRT, the only enzyme known to metabolize NA in mammals. Strikingly, *Naprt* knock-out mice

showed robustly decreased blood NAR and increased NA (Fig. 7e, f and Supplementary Fig. 9d). We next asked if the decrease in circulating NAR seen in *Naprt*-KO mice altered NAMPTi therapeutic efficacy. For this purpose, we analyzed the A2780 ovarian cancer line, as those cells are (1) highly susceptible to NAMPT inhibition[32] and can use NAR, but not NA, for NAD synthesis in culture (Supplementary Fig. 9e) and (2) transplantable into *Rag-1*-KO mice. GNE-617 treatment had a stronger tumor-shrinking effect in *Naprt*-KO mice harboring A2780 tumors than in comparable *Naprt*-WT mice (Fig. 7g and Supplementary Fig. 9f). Collectively, these data indicate that circulating NAR is derived from dietary NA NAPRT-dependently

**Fig. 3 | Absence of QPRT-dependent de novo NAD synthesis promotes NAMPTi susceptibility in SCLC. a, b** Relative NAD (**a**) and ATP (**b**) levels in 7 NSCLC and 4 SCLC cell lines treated 48 h with NAMPTi. Results are shown as values relative to those seen in non-treated controls. Each symbol represents an individual cell line. $P = 0.0022$ (**b**). **c** Adenylate and guanylate energy charge of cells treated 48 h with or without NAMPTi. Shown are results of 7 NSCLC and 4 SCLC cell lines. $P = 0.0789$ (SCLC, adenylate); $P = 0.0215$ (SCLC, guanylate). **d** NAD de novo synthesis pathway. Kyn Kynureine, 3OH-Anthr 3-OH-anthranilic acid, QA quinolinic acid. **e** Expression of genes encoding enzymes shown in (**d**) in 50 SCLC and 135 NSCLC cell lines collected in the DepMap dataset. **f** mRNA levels of three indicated genes whose expression significantly differs between SCLC and NSCLC groups. $P = 0.0006$ (*TDO2*); $P = 0.0012$ (*KMO*); $P < 0.0001$ (*KYNU*). **g** Immunoassays showing knock-down efficiency of indicated siRNAs in H1975 NSCLC cells. The blot shown is representative of three independent experiments. **h** Effects of knockdown of either *KYNU* or *QPRT* on NAD levels in H1975 cells. $n = 3$ biological replicates. $P = 0.0044$ (NTC vs KYNU, wo FK866); $P = 0.0001$ (NTC vs QPRT, wo FK866); $P = 0.0036$ (NTC vs QPRT, w/ FK866). **i, j** Immunoassays (**i**) and NAD levels (**j**) of SCLC cells transduced with Flag-QPRT or empty vector (EV). The blot shown is representative of two independent experiments (**i**). $n = 3$ (87-5/EV and H209/EV cells) or 4 (87-5/QPRT and H209/QPRT cells) biological replicates in (**j**). $P < 0.0001$ (EV vs QPRT, wo FK866, 87-5); $P < 0.0001$ (EV vs QPRT, w/ FK866, 87-5); $P < 0.0001$ (EV vs QPRT, wo FK866, H209); $P < 0.0001$ (EV vs QPRT, w/ FK866, H209). **k** SCLC cells in i were treated 4 days with 20 nM FK866 and viable cells were scored. Dashed lines represent cell number at day 0. $n = 3$ (87-5/EV, 87-5/QPRT and FK866-treated H209/EV or H209/QPRT cells) or 4 (untreated H209/EV or H209/QPRT cells) biological replicates. $P < 0.0001$ (EV vs QPRT, w/ FK866, 87-5); $P < 0.0001$ (EV vs QPRT, w/ FK866, H209). Data are presented as mean plus SEM. $*P < 0.05$, $**P < 0.01$, $***P < 0.001$, $****P < 0.0001$ as determined by one-way ANOVA with a post hoc test (**c, h, k**) or by two-tailed $t$ test (**a, b, f**). Source data are provided as a Source Data file.

and that the NA-to-NAR conversion may limit the therapeutic anti-tumor efficacy of NAMPTi (Fig. 7h).

## Discussion

Our findings demonstrate the vulnerability of lung and prostate NECs to inhibition of NAD salvage. These results are consistent with previous reports suggesting high vulnerability of either YAP-negative or a subset of small-cell cancers to NAMPTi[33,34]. SCLCs, SCPCs and CRPCs harbor few druggable driver mutations, and since progress in developing targeted therapies has been minimal, the metabolic vulnerability of these cancers may provide unique therapeutic opportunities. Given that high NAMPT dependence is associated with NE differentiation, our findings are also likely applicable to NEC(s) of organs other than lung and prostate. Previous studies also report that some AML leukemia cells show high NAMPT dependence, although mechanisms underlying death of those cells after NAD depletion by NAMPT inhibition likely differ from those seen in NECs[5,6]. Whether the effects of NAMPTi on AML are altered by dietary niacin is now an important question. Also, some non-NEC and non-AML lines show vulnerability to NAMPTi comparable to that shown by A2780 cells (Fig. 7g and Supplementary Fig. 9e). Thus, some sporadic cases may be vulnerable to NAMPT inhibition, regardless of cancer type. Determining the molecular basis of that vulnerability is a question for future study.

We observed significant differences in expression of genes relevant to NAD de novo synthesis, such as *KYNU*, in SCLC compared to NSCLC, and RNAi analysis confirmed that QPRT-dependent NAD synthesis underlies in part NAMPTi-resistance in NSCLC (Fig. 3). However, we found that residual NAD levels in QPRT-silenced cells remained higher than those seen in SCLC after NAMPTi treatment. The reason for these outcomes is unclear, although SCLC and NSCLC cells likely exhibit as yet uncharacterized differences in NAD metabolism, such as differences in NAD consumption rates. Further study is needed to assess these possibilities.

This study also shows that dietary NA is converted to circulating NAR. Our results challenge a previous model emphasizing the importance of circulating NA and NR in antagonizing NAMPT-targeting cancer therapies[27,28,35]. Instead, we propose that circulating NAR may compensate for NAMPT inactivation in NAMPTi-treated tumor cells, particularly in the absence of de novo NAD synthesis. Another important finding reported here is that blood NAR levels are readily lowered by restriction of dietary NA and that NA-restriction may robustly boost the therapeutic efficacy of NAMPT inhibition. These findings illustrate the power of nutritional approaches to cancer therapy, a topic of recent interest[20,21]. Moreover, since NAR metabolism requires NMRK1, treatment with a small molecule inhibitor of tumor cell NMRK1 could serve as another option to drive synthetic lethality in the presence of a NAMPTi. By contrast, inhibiting NADSYN1 may be toxic as it likely inhibits NAD and Nam synthesis from Trp in liver[25]. NADSYN1

inactivation may be partially modeled in mice fed a WFD, which is toxic, as shown in Supplementary Fig. 4a. Blood Nam decreased in mice fed a WFD (Supplementary Fig. 7f) likely due to poor conversion of Trp to Nam in the host. A WFD also lowered blood NAR levels, although the mechanism remains unknown. Lowering of Nam and/or NAR may, in turn, impair NAD synthesis in tumor cells and synergize with NAMPTi treatment (Fig. 5a and Supplementary Fig. 4b, d). Lower blood Nam may also lead to WFD's toxicity since a NFD, which decreases NAR but not Nam, was tolerated by mice even when combined with a NAMPTi. Further study is needed to address these issues.

The observation that NFD's effect on circulating niacin is selective for NAR was unanticipated. Unlike NAR, blood levels of NA and NR were almost unchanged in the presence of a NFD (Fig. 6c). In vitro, NA and NR can rescue cells from the toxicity of NAMPTi-induced NAD depletion, but only at supra-physiological concentrations (compare results shown in Fig. 6c and Supplementary Fig. 7a). Thus, here, we found no evidence that tumor cells could benefit from their ability to use NA or NR for NAD synthesis, particularly in vivo. However, we do not exclude the possibility that in some tumor microenvironments, local concentrations of NA and/or NR are high enough to promote NAD synthesis in tumor cells. Since NAD is required for many physiological tissue or cellular functions, altering NAD metabolism may impact both tumor and non-tumor tissues. Thus, possible adverse effects on healthy tissues/cells should be carefully examined before translating our results into a clinical setting. Although mice tolerated a combination of NAMPTi treatment and restriction of dietary niacin, we note that humans and rodents may exhibit difference(s) in niacin/NAD metabolism. Nevertheless, our study reports a previously unrecognized role of NAR in cancer biology.

## Methods

All experiments using human organoids TR-6TF were performed with approval of the ethics committees at Miyagi Cancer Center and Keio University. TR-6TF was established by genetic manipulation of an organoid line established from non-tumor colon tissue from a female colon cancer patient in her 70's after obtaining informed consent in a previous study[14]. All animal experiments were performed with approval of the Miyagi Cancer Center Research Institute Animal Care and Use Committee.

### Reagents

Nicotinamide (Nam), nicotinic acid (NA) and nicotinamide riboside, [$^{15}$N(amide)]-Nam were purchased from Merck (Darmstadt, Germany). Nicotinic acid riboside was purchased from Toronto Research Chemicals (Toronto, Canada) or BLDPharm (Shanghai, China). [$^{13}$C-U]-glucose and [D4]-NA were purchased from Cambridge Isotope Laboratories Inc (Tewksbury, MA). FK866 was from Cayman (Ann Arbor, MI). GNE-617 was purchased from Med Chem Express

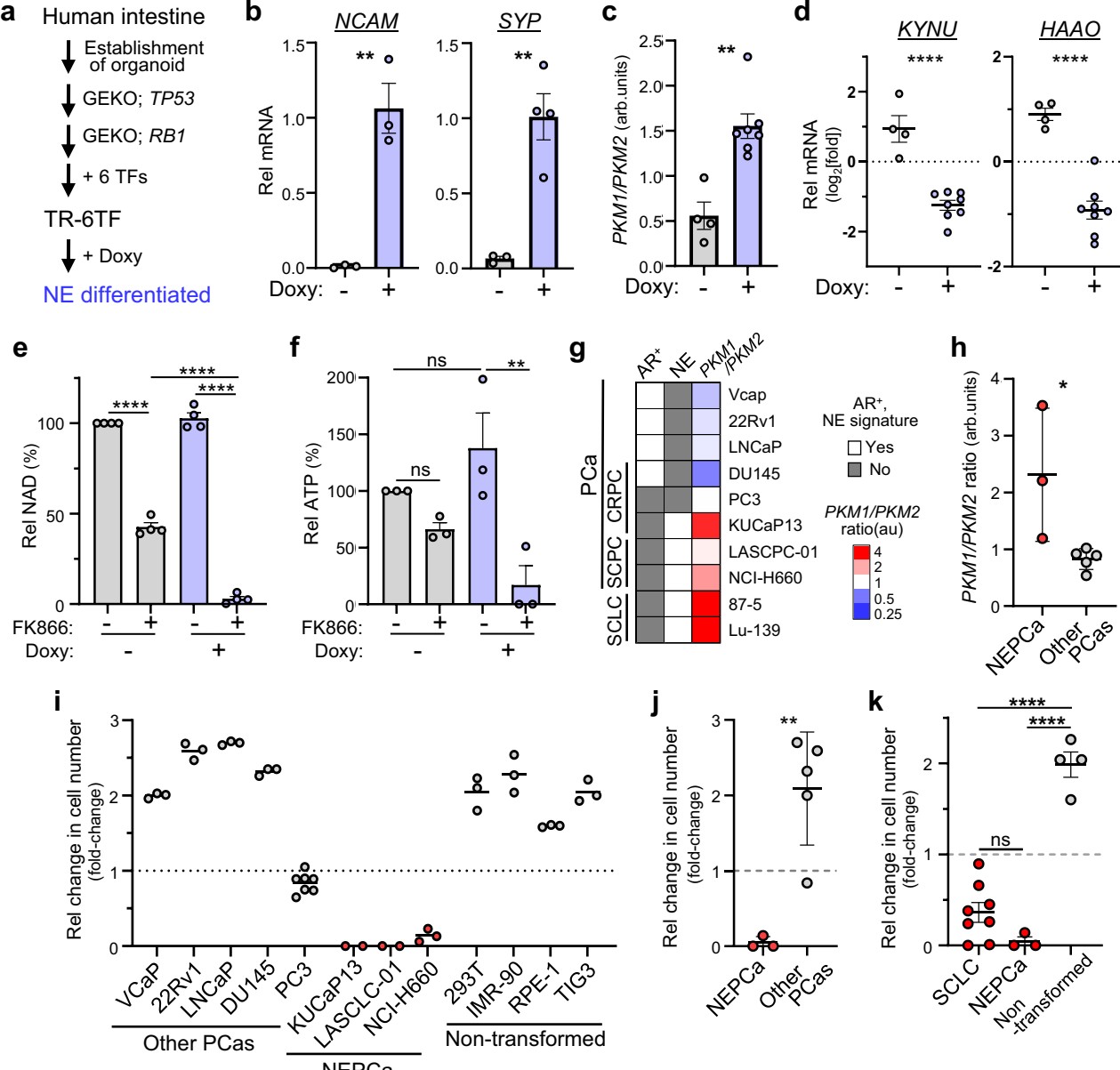

**Fig. 4 | High NAMPT-dependence is associated with NE phenotypes. a** Schematic showing establishment of TR-6TF, a human organoid that can be differentiated into a NE lineage following doxycycline (Doxy)-dependent induction of 6 transcription factors (TF). 6TF: ASCL1, NEUROD1, NKX2-5, POU3F2, SOX2 and TP73. GEKO, gene knock-out. **b, c** qRT-PCR analyses of TR-6TF organoids. Shown are levels of transcripts encoding NE markers (**b**) and the *PKM1/PKM2* mRNA ratio (**c**). $n = 4$ (Doxy-treated group for *SYP*) or 3 (all others) biological replicates in (**b**). $n = 4$ (untreated) or 7 (Doxy-treated) biological replicates in (**c**). $P = 0.0032$ (*NCAM*); $P = 0.0036$ (*SYP*) (**b**). $P = 0.0012$ (**c**). **d** qRT-PCR analyses of TR-6TF organoids. Shown are *KYNU* and *HAAO* transcript levels. $n = 4$ (untreated) or 8 (Doxy-treated) biological replicates. $P < 0.0001$ (*KYNU*); $P < 0.0001$ (*HAAO*). **e, f** NAD (**e**) and ATP (**f**) levels in TR-6TF organoids cultured in the presence or absence of Doxy and then treated or untreated 48 h with FK866. $n = 3$ or 4 biological replicates. $P < 0.0001$ (w/ vs wo FK866, wo Doxy); $P < 0.0001$ (wo vs w/ Doxy, w/ FK866); $P < 0.0001$ (wo vs w/ FK866, w/ Doxy) (**e**). $P = 0.5724$ (wo vs w/ FK866, wo Doxy); $P = 0.4892$ (wo vs w/ Doxy, wo FK866); $P = 0.0064$ (wo vs w/ FK866, w/ Doxy) (**f**). **g** Classification of prostate cancer (PCa) lines divided into groups based on histopathological types

(SCPC, CRPC, or not), androgen receptor (AR) expression and a neuroendocrine (NE) signature. The *PKM1/PKM2* ratio was determined by qRT-PCR analysis. Scores are averages of at least two duplicates. Two SCLC lines at the bottom (87-5 and Lu-139) are shown for comparison. **h** Calculation of the *PKM1/PKM2* ratio in NE-type (NEPCa) versus other PCas. Shown are results of 3 NEPCa and 5 other PCa cell lines. $P = 0.0265$. **i** Fold-change in the number of cells in PCa and non-transformed lines cultured 4 days with 20 nM FK866. Dashed line represents cell number at day 0. $n = 2$ (KUCaP13 and LASCLC-01), 7 (PC-3) or 3 (all others) biological replicates. **j** Comparison of results shown in (**i**) between NEPCa and other PCas. Shown are results of 3 NEPCa and 5 other PCa cell lines. $P = 0.0038$. **k** Comparison of changes in cell number after 4 days of FK866 treatment in SCLC (results shown in Fig. 2c), NEPCa and non-transformed lines (**i**). Shown are results from 8 SCLC, 3 NEPCa and 4 non-transformed cell lines. $P = 0.2459$ (SCLC vs NEPCa); $P < 0.0001$ (SCLC vs non-transformed); $P < 0.0001$ (NEPCa vs non-transformed). Data are presented as mean plus the SEM. *$P < 0.05$, **$P < 0.01$, ****$P < 0.0001$ as determined by one-way ANOVA with a post hoc test (**e, f, k**) or by two-tailed *t* test (**b, c, d, h, j**). ns., not significant. Source data are provided as a Source Data file.

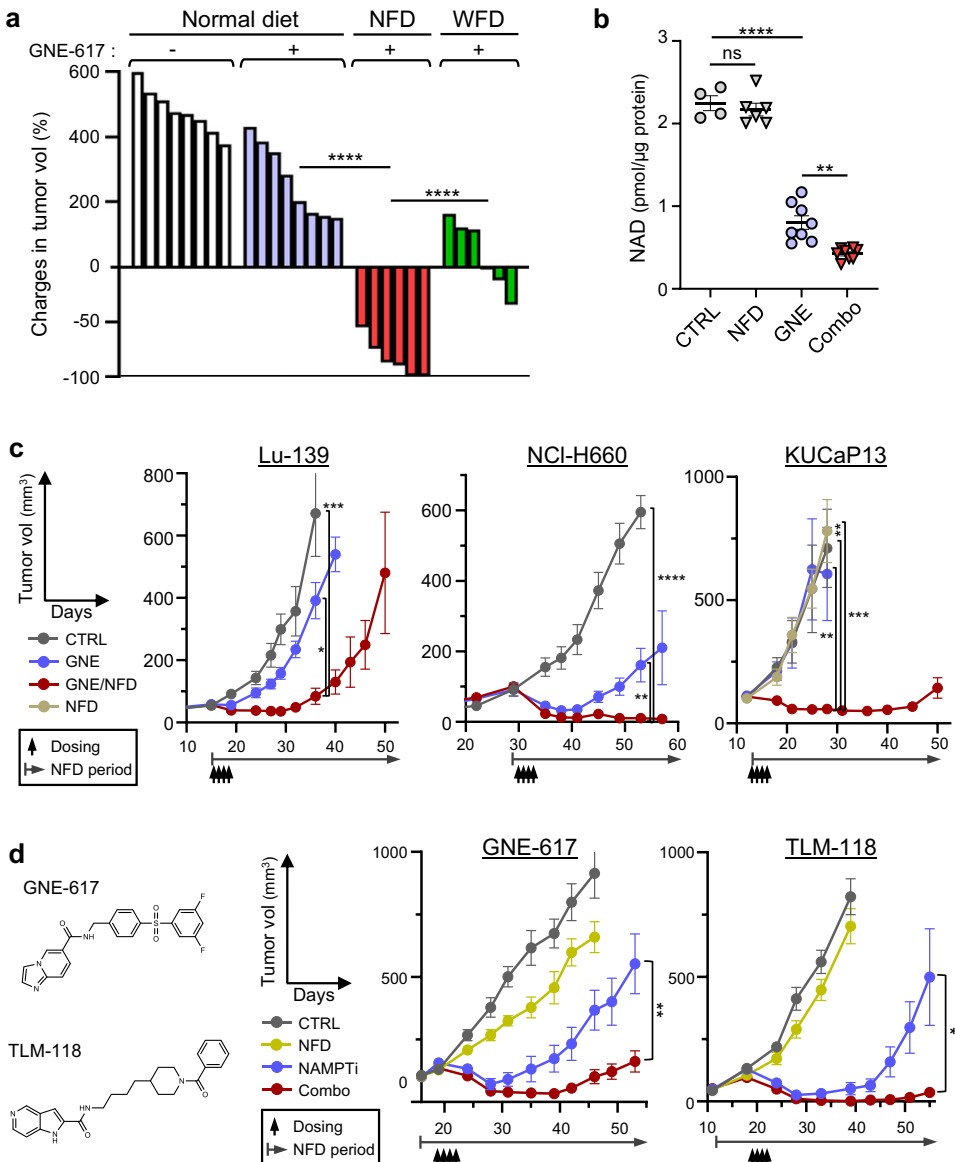

**Fig. 5 | Restriction of dietary niacin enhances NAMPT-targeting therapy in mice. a** Effects of either NFD or WFD on GNE-617 (GNE) therapeutic effects. Volumes of Lu-139 tumors at day 12 are shown relative to baseline (day 0) volumes. Each bar represents a tumor. *n* = 8 (Normal diet groups) or 6 (NFD and WFD) tumors. *P*<0.0001 (Normal diet vs NFD, GNE-617); *P*<0.0001 (NFD vs WFD). **b** NAD levels in Lu-139 tumors from mice treated with NFD, NAMPTi (GNE) or both (Combo). Each symbol represents one tumor. *n* = 4 (CTRL), 6 (NFD) or 8 (all others) tumors. *P* = 0.9034 (CTRL vs NFD); *P*<0.0001 (CTRL vs GNE); *P* = 0.0016 (GNE vs Combo). **c** Growth curves of SCLC (Lu-139), SCPC (NCI-H660) and NEPCa (KUCaP13) xenograft tumors in mice treated with vehicle (CTRL), GNE-617 alone (GNE) or GNE-617 plus NFD (Combo). KUCaP13 graph also includes results from mice treated only with NFD. *n* = 6 (CTRL and GNE) or 8 (GNE/NFD) Lu-139 tumors; *n* = 6 (CTRL), 5 (GNE) or 9 (GNE/NFD) NCI-H660 tumors; *n* = 4 (CTRL), 6 (GNE and NFD) or 10 (GNE/NFD) KUCaP13 tumors. *P* = 0.0002 (CTRL vs GNE/NFD, day 36, Lu-

139); *P* = 0.0335 (GNE vs GNE/NFD, day 36, Lu-139); *P* <0.0001 (CTRL vs GNE/NFD, day 53, NCI-H660); *P* = 0.0065 (GNE vs GNE/NFD, day 53, NCI-H660); *P* = 0.0053 (CTRL vs GNE/NFD, day 28, KUCaP13); *P* = 0.0005 (NFD vs GNE/NFD, day 28, KUCaP13); *P* = 0.0074 (GNE vs GNE/NFD, day 28, KUCaP13). **d** Analyses using two structurally dissimilar NAMPT inhibitors (GNE-617 and TLM-118). (left) Compound structures. (right panels) Growth curves of 87-5 SCLC tumors in mice treated with NFD, a NAMPTi (GNE-617 or TLM-118) or a combination of NFD + the indicated NAMPTi. *n* = 10 (CTRL, NAMPTi), 7 (NFD) or 12 (Combo) tumors in GNE-617 experiments; *n* = 8 tumors in TLN-118 experiments. *P* = 0.0029 (NAMPTi vs Combo, day 53, GNE-617); *P* = 0.0364 (NAMPTi vs Combo, day 56, TLM-118). Data are presented as mean plus the SEM. *\*P*<0.05, \*\**P*<0.01, \*\*\**P*<0.001, \*\*\*\**P*<0.0001 as determined by one-way ANOVA with a post hoc test (**a**–**c**) or by two-tailed *t* test (**d**). ns. not significant. Source data are provided as a Source Data file.

(Monmouth Junction, NJ). Koningic acid was purchased from Biolinks K.K. (Tokyo, Japan). Synthesis of TLM-118 is reported elsewhere[26].

## Human and mouse cell lines

Human cell lines Lu-134 (RCB0466), Lu-139 (RCB0469), Lu-165 (RCB1184), NCI-H209 (HTB-172), T3M-12 (RCB2281), 87-5 (RCB2092), MS-1 (RCB0725), S1 (RCB1966), ABC-1 (JCRB0815),

H1975 (CRL-5908), H1299 (CRL-5803), A549 (RCB0098), LC-KJ (JCRB0137), EBC-1 (JCRB0820), LK-2 (JCRB0829), VCaP (CRL-2876), 22Rv1 (CRL-2505), DU145 (RCB2143), LNCaP (RCB2144), PC-3 (CRL-1435), NCI-H660 (CRL-5813), LASCPC-01 (CRL-3356), 293T (RCB2202), IMR-90 (JCRB9054), RPE-1 (CRL-4000), TIG3 (JCRB0506), NOMO-1 (IFO50474), THP-1 (JCRB0112.1), MOLM-14 (JCRB1812) and U937 (JCRB9021) were provided by the Riken

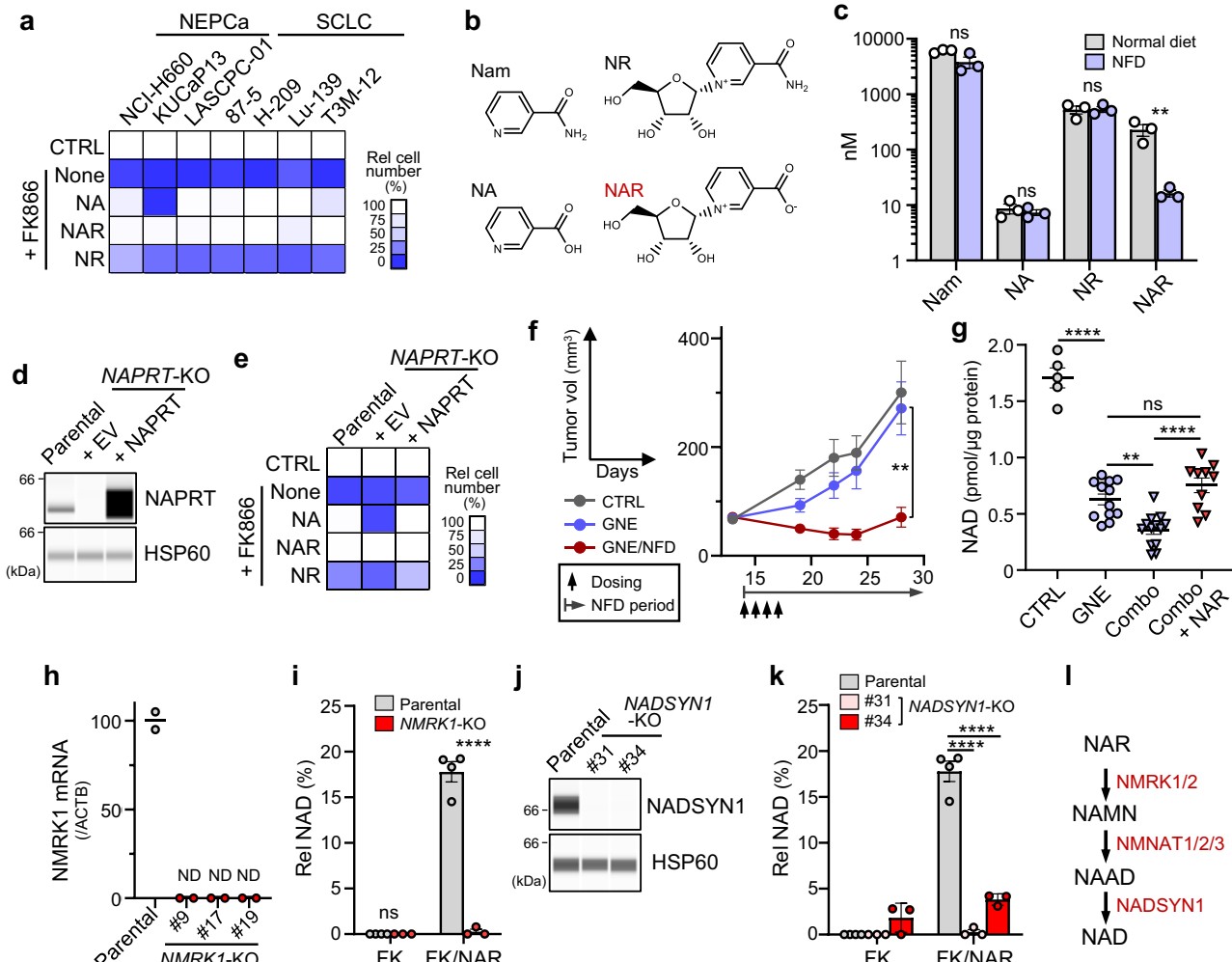

**Fig. 6 | Niacin restriction lowers blood NAR and restricts NAMPT-independent NAD synthesis in tumor cells. a** Effects of 10 μM exogenous niacin (NA, NAR or NR) on growth suppression by FK866 in indicated cultured cells. **b** Structures of niacin molecules found in mouse serum. **c** Serum levels of niacin molecules shown in (**b**) in mice fed a normal diet or NFD for 3 days. *n* = 3 mice (both groups). *P* = 0.0721 (Nam); *P* = 0.5360 (NA); *P* = 0.9509 (NR); *P* = 0.0180 (NAR). **d** *NAPRT* knock-out and rescue. *NAPRT* was knocked out in 87-5 cells, and one KO clone was transduced with either an NAPRT expression vector or empty vector (EV). The blot shown is representative of >2 independent experiments. **e** Effects of exogenous niacin on growth suppression by FK866 in cells or the clone in (**d**). **f** Growth curves of *NAPRT*-KO 87-5 xenograft tumors treated as in Fig. 5. *n* = 6 (CTRL) or 8 (GNE and GNE/NFD) tumors. *P* = 0.0067 (GNE vs GNE/NFD, day 28). **g** NAD levels of Lu-139 tumors in mice treated either as in Fig. 5 or with Combo + NAR. Combo: "GNE + NFD". Each symbol represents one tumor. *n* = 5 (CTRL), 12 (GNE), 13 (Combo) or 10 (Combo + NAR) tumors. *P* < 0.0001 (CTRL vs GNE); *P* = 0.0022 (GNE vs combo); *P* =

0.3450 (GNE vs Combo + NAR); *P* <0.0001 (Combo vs Combo + NAR). **h** qRT-PCR analyses of *NMRK1* in parental and *NMRK1* knock-out 87-5 clones. Shown are mean values from two duplicate measures. ND not detectable. **i** NAD levels in parental and *NMRK1*-KO cells. Parental cells were assessed as four biological replicates. KO clones, each represented by a symbol, are the three independent clones shown in (**h**). FK, FK866. *P*<0.0001 (Parental vs NMRK1-KO, FK/NAR). **j** Immunoassays of *NADSYN1*-KO 87-5 clones. Shown is representative analysis of two assays. **k** NAD levels in *NADSYN1*-KO cells. Parental cell levels are the same as those shown in (**i**). Also shown are results of two independent KO clones, each with three biological replicates. *P* <0.0001 (Parental vs #31, FK/NAR); *P*<0.0001 (Parental vs #34, FK/NAR). **l** NAD synthesis pathway from NAR. Data are presented as the mean plus SEM (**c, f, g, i, k**). **P*<0.01, *****P*<0.0001 as determined by one-way ANOVA with a post hoc test (**g, k**) or by two-tailed *t* test (**c, f, i**). ns not significant. Source data are provided as a Source Data file.

Bioresource Center (Tsukuba, Japan), JCRB (Tokyo, Japan) or ATCC (Manassas, VA). The human ovarian cancer line A2780 was kindly provided by Dr Carla Grandori. The human NSCLC cell line HS24 was kindly provided by Dr Makoto Maemondo. The human CRPC line KUCaP13[19] was cultured in RPMI1640 medium supplemented with 10% FCS, Pen/Strep/Glutamine solution and 12.5 mM HEPES. NCI-H660 cells were cultured in HITES medium. Other lines were cultured in RPMI supplemented with 10% FCS. MEFs and LE cells were established from *Pkm*-knock-in mice (*Pkm*[M1/M1] and *Pkm*[M2]), transformed with oncogenic Kras, and cultured as described[4,36]. Mutant mouse strains were deposited with the RIKEN BioResource Research Center (Tsukuba, Japan). Three or four independent MEF and LE lines

derived from 3 or 4 independent mice per genotype were used in this study.

**Culture analysis of cell lines and organoids**
Medium for [15]N-Nam labeling was prepared by adding 4 mg/L (32.8 μM) [15]N Nam, 10% FCS (dialyzed), 16 mg/L Trp, 1 mM Na-pyruvate, 2 g/L glucose and Pen/Strep/Glutamine solution to DMEM free of Nam, Trp Na-pyruvate, glucose, and glutamine (IFP Co Ltd, Higashine, Japan). For [13]C-glucose labeling of mouse cells, cells were incubated with DMEM containing [U-13C] glucose at 1000 mg/L and processed as described[4]. For tracer experiments in human lines, medium was prepared by adding 10% FCS (dialyzed) and 2000 mg/L

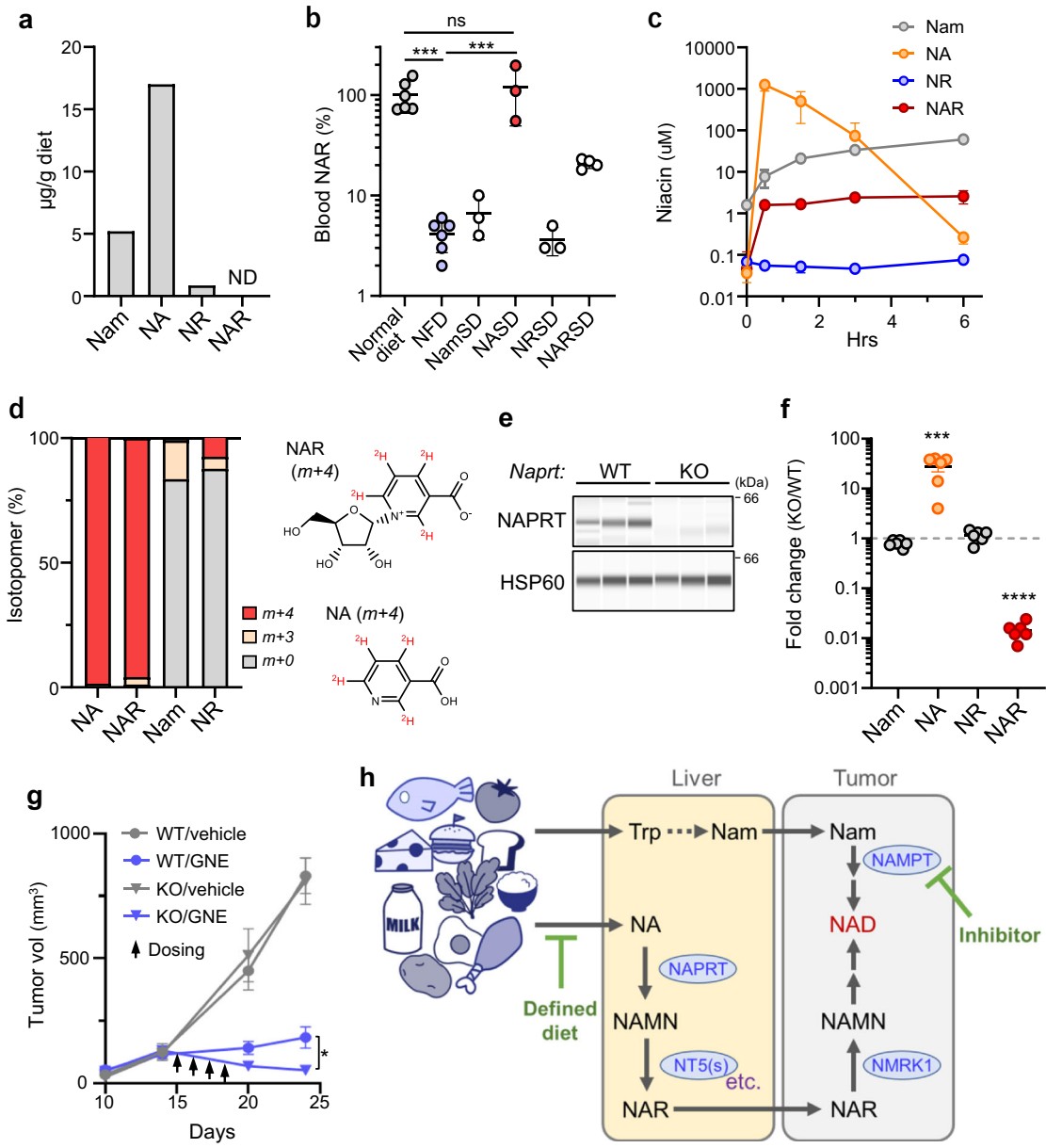

**Fig. 7 | Blood NAR is derived from dietary NA, NAPRT-dependently. a** Amounts of indicated niacin molecules found in the normal diet used for mice experiments. ND not detectable. Shown are averages of >2 technical replicates. **b** Effects of niacin supplementation of NFD on mouse serum NAR levels. *n* = 6 (normal diet and NFD), 4 (NARSD) or 3 (all others) mice. *P* = 0.0002 (Normal diet vs NFD); *P* = 0.9188 (Normal diet vs NASD); *P* = 0.0002 (NFD vs NASD). **c** Effects of NA administration (by gavage) on serum niacin levels. *n* = 3 (0, 1.5 and 6 h) or 4 (0.5 and 3 h) mice per time point. **d** Levels of labeled and unlabeled niacin molecules in serum of mice administered deuterium-labeled NA ([D4]-NA). Data is from an average of five mice. NAR (m+4) structure derived from [D4]-NA (m+4) is also shown. **e** NAPRT expression in liver of WT and *Naprt*-KO mice. *n* = 3 mice for each genotype. **f** Effects of

*Naprt*-deficiency on serum levels of niacin molecules. *n* = 6 mice per genotype. *P* = 0.0003 (NA); *P* <0.0001 (NAR). **g** Effects of host *Naprt*-deficiency on NAMPTi therapeutic efficacy. *Naprt*WT;*Rag1*-/- or *Naprt*-/-;*Rag1*-/- mice were inoculated with A2780 cells and treated with GNE-617. *n* = 6 (WT/vehicle KO/vehicle), 7 (KO/GNE) or 8 (WT/GNE) tumors. *P* = 0.0136 (WT/GNE vs KO/GNE, day 24). **h** Proposed model of how systemic niacin metabolism modulates efficacy of NAMPT-targeted therapy. Data are presented as mean (**d**) or the mean plus SEM (**b, c, f, g**). *P <0.05, ***P < 0.001, ****P <0.0001 as determined by one-way ANOVA with a post hoc test (**b**) or by two-tailed *t* test (**f, g**). ns not significant. Source data are provided as a Source Data file.

[$^{13}$C-U] glucose to glucose-free RPMI medium (FUJIFILM Wako, Osaka, Japan).

FK866 was used at 20 nM. For niacin rescue experiments, cells were simultaneously treated with FK866 and 10 μM NA, NAR or NR, unless otherwise specified. For GAPDH inhibition experiments in SCLC, PBS/EDTA-treated cells were resuspended in RPMI/FCS medium, and then 5 μM Koningic acid (KA) (or control DMSO) was added. Alternatively, medium was replaced with new RPMI/FCS medium

containing 5 μM KA for adherent cells (as performed for all NSCLC cell lines used in this paper).

TR-6TF organoids were maintained as standard Matrigel-embedded 3D culture. Organoids were suspended in 25 μl of ice-cold Matrigel (growth factor-reduced, Corning, Glendale, AZ), plated on 48-well plates, and left 10 min at 37 °C to allow gel formation. Then, 0.25 ml of medium was added to wells. Basal medium was defined as follows: Advanced DMEM/F12 (Thermo Fisher, Waltham, MA)

supplemented with penicillin/streptomycin, 10 mM HEPES, 2 mM GlutaMAX, B27 (Thermo Fisher), 10 nM gastrin I (Merck), and 1 mM N-acetylcysteine (FUJIFILM Wako). Complete medium was prepared by supplementing basal medium with the following niche factors: 50 ng/ml mouse recombinant EGF, 50 ng/ml human recombinant FGF-2, 100 ng/ml mouse recombinant noggin (Thermo Fisher), 100 ng/ml human recombinant IGF-1 (BioLegend, San Diego, CA), 1 µg/ml recombinant human R-spondin-1 (R&D, Minneapolis, MN), 10% Afamin-Wnt-3A conditioned medium (MBL, Nagoya, Japan), and 500 nM A83-01 (Tocris, Bristol, UK). NE-differentiation was induced by adding 100 ng/ml doxycycline to complete medium lacking Wnt-3A and R-spondin-1.

### RNA interference
ON-target plus SMARTpool siRNAs, which consist of a mixture of 4 siRNAs designed to silence a single target−either human NAMPT, NAPRT, QPRT, NADSYN1, NMRK1 or KYNU−were obtained from Horizon Discovery (Cambridge, UK). An ON-target plus non-targeting control pool (Horizon discovery) served as control siRNA. Two additional NAMPT siRNAs (#2 and #3, both from Thermo Fisher) were used to verify specificity of NAMPT knockdown effects, as shown in Supplementary Fig. 1b. siRNAs were transfected using LipofectAMINE RNAiMAX (Thermo Fisher) according to the manufacturer's recommendation. In experiments shown in Fig. 2b and Supplementary Fig. 1a–c, cells were harvested 3 days after transfection.

### Genome editing
Recombinant Cas9 proteins (True Cut Cas9 Protein V2) and sgRNA targeting human NAPRT, NMRK1 and NADSYN1 gene were purchased from Thermo Fisher Scientific. Genome editing was performed by transfection of RNP complexes consisting of Cas9 protein and sgRNAs. 87-5 and Lu-139 cells were transfected using LipofectAMINE CRISPRMAX reagent (Thermo Fisher), following the manufacturer's recommendations. *NMRK1-* or *NADSYN1*-KO clones were obtained by genome editing of a *NAPRT*-KO 87-5 clone. Transfection of the *NAPRT*-KO 87-5 clone was performed by electroporation using a NEPA gene electroporator (Nepa Gene, Ichikawa, Japan) at the following settings: the poring-pulse was set for 175 V with a 5 msec pulse-width, and the transfer-pulse was set for 20 V with a 50 msec pulse-width. RNP-transfected cells were subjected to a cloning procedure using the standard limiting dilution method. KO clones were selected based on loss of either NAPRT or NADSYN1 protein or loss of NMRK1 mRNA.

### Lentivirus-mediated gene expression
The lentivirus plasmids pLV-EFS-HA/NAPRT-Neo, pLV-EFS-Flag/QPRT-Neo and appropriate empty vector were sourced from VectorBuilder (Chicago, IL). Lentivirus was produced by standard procedures using 293T cells, psPAX2 and pMD2.G packaging plasmids, and lentivirus plasmids. Virus infections were performed using a 3:1 mixture of 293T culture supernatant and fresh medium. Polybrene was added to the virus mixture at 8 µg/ml to promote infection. Infected cells and organoids were selected in 500 µg/ml Geneticin.

### Analysis of the Cancer Dependency Map (DepMap) dataset
The dataset published in the release of "DepMap Public 22Q2" (https://depmap.org/portal/ download/)[9,10] was used to assess the impact of *NAMPT* knock-out by genome-editing on proliferation of cell lines in the CCLE collection. The sgNRA library used in the DepMap project was reported to contain four sgRNA barcodes targeting the *NAMPT* gene[10]. The same dataset was used to analyze mRNA expression. Histological classification of cell lines was according to the DepMap portal or Expasy operated by the BROAD Institute or the Swiss Institute of Bioinformatics, respectively.

### qRT-PCR analyses
Total RNAs were reverse-transcribed using random primers and Superscript III RTase (Thermo Fisher). qRT-PCR analyses were performed using a LightCycler 480 (Roche, Basel, Switzerland). LightCycler 480 probes master and TaqMan probes from a universal probe library (Roche) were used for analyses of *ACTB* (#64), *CHGA* (#17), *GAPDH* (#9), *NCAM* (#55), *NMRK1* (#75), *NMRK2* (#60), *PKM1* (#46), *PKM2* (#13), *QPRT* (#49), *RN18s* (#48), *SYP* (#13) and *YAP* (#59) (numbers in parentheses indicate probe number). Primers used are listed in Supplementary Data 1. Analyses of *ACMSD*, *AFMID*, *HAAO*, *IDO1*, *IDO2*, *KMO*, *KYNU* and *TDO2* were performed using a TaqMan Gene Expression Assay kit (Thermo Fisher), as indicated in Supplementary Data 1.

### Western blot analysis and capillary-based immunoassays
Conventional Western blot analysis was performed using TXN gels and a Transblot Turbo blotting system (BioRad, Hercules, CA). Alternatively, proteins were size-separated and immune-detected using an automated capillary-based immunoassay system (JESS Simple Western system from ProteinSimple, San Jose, CA). Antibodies used were Anti-PAR mAb (#10407, IBL, Fujioka, Japan, 1:200), anti-Actin mAb (A3853, Thermo Fisher, 1:500), anti-NAMPT (ab45890, Abcam, Cambridge, UK, 1:500), anti-NAPRT (ab127699, Abcam, 1:2000), anti-NADSYN1 (ab171942, Abcam, 1:900), anti-QPRT (ab171944, Abcam, 1:1000), anti-HSP60 (AF1800, ProteinSimple, 1:500), anti-KYNU (11796-1-AP, ProteinTech, Rosemont, IL, 1:200) and anti-HAAO (12791-1-AP, ProteinTech, 1:300). Anti-mouse IgG-HRP (sc-2055, Santa Cruz, 1:3000) and anti-rabbit IgG-HRP (5220-0336, SeraCare, Milford, MA, 1:3000) served as secondary antibodies in Western blotting. Anti-mouse Ig-HRP (042-205, Protein Simple, no dilution), anti-rabbit Ig-HRP (042-206, Protein Simple, no dilution) and anti-mouse Ig-NIR (043-821, Protein Simple, 1:200) were used as secondary antibodies for the JESS system. Uncropped images of each blot are provided in the Source Data file.

### Metabolite measurement by colorimetric/fluorometric probes
An ATP Colorimetric/Fluorometric Assay kit and a NAD+/NADH Quantification kit were purchased from BioVision (Waltham, MA) and used to quantify respective cellular ATP and NAD levels. Cells were collected and lysed, and protein concentrations in lysates were determined by a BCA protein assay (Thermo Fisher) and used to normalize metabolite levels.

### Metabolite measurement by mass spectrometry
To analyze cellular metabolite levels, cells treated or untreated with FK866 were washed with ice-cold PBS, snap-frozen in liquid nitrogen and stored at -80 °C until use. Similarly, tissue samples were washed with PBS and snap-frozen immediately after resection. For blood samples, whole blood samples collected from anesthetized mice were immediately subjected to serum isolation and then snap-frozen.

Water soluble metabolites were extracted from cells or tissues with the standard Bligh-Dyer method using $H_2O$, methanol and chloroform. Resulting extracts were evaporated and reconstituted in $H_2O$ (for Liquid Chromatograph (LC)-mass spectrometry (MS) system) or $H_2O$ containing 200 µM 3-aminopyrrolidine and 200 µM trimesate (for capillary electrophoresis (CE)-MS system).

For global metabolome analysis, samples were analyzed by CE-MS using the Agilent CE capillary electrophoresis system, a G6230B LC/MSD TOF system, a 1260 Infinity 2 series binary HPLC pump, and the G1603A Agilent CE-MS adapter- and G1607A Agilent CE-ESI-MS sprayer kit (all from Agilent Technologies, Santa Clara, CA). For niacin analyses, samples were analyzed by LC-MS either using an Agilent 6460 Triple Quad mass spectrometer coupled with an Agilent 1290 HPLC system (Agilent) or QTRAP 5500 (AB SCIEX LLC, Framingham, MA) coupled with Nexera XR (Shimadzu, Kyoto, Japan). In analysis of niacin in

mouse serum, we evaluated the following metabolites/nutrients: NAMN, NMN, cADPR, ADPR, NAAD, NAD, Nam, NR, NA, and NAR. All but the last four were below detection limits. Energy charge was calculated using the following formulation: energy charge = ([ATP] + 0.5x[ADP])/([ATP]+[ADP]+[AMP]).

## Mouse experiments

Mice were maintained in specific pathogen-free (SPF) facilities with a 12-h light–dark cycle, controlled temperature of ~25 °C, and controlled humidity of ~50%. The maximal tumor size/burden permitted by our institutional review board is 10% of body weight (combined burden if more than one mass present), which corresponds to a tumor volume of 2200 mm$^3$ in adult mice (~22 g). The maximal tumor size/burden permitted by our institutional review board was not exceeded in any transplantation experiment. Unless stated, mice were fed a normal diet (MF from Oriental Yeast Co., LTD; Tokyo, Japan) ad libitum.

Xenografts made using human cell lines were generated by s.c. inoculation of 8–12 week-old female NOG mice (In-Vivo Science, Tokyo, Japan) with 1 x 10$^6$ cells plus Matrigel per site. Tumor length and width were measured by calipers, and volume was calculated based on the standard formula: (length × width$^2$)/2. GNE-617 was dissolved at 3 mg/ml in PEG400/H$_2$O/EtOH (6:3:1) solution by sonication, and dosed at 0.2 ml/mouse by gavage once a day at 12 p.m. for 4 consecutive days. TLM-118 was dissolved at 6 mg/ml in Cremophor/EtOH/ PBS (1:1:4) solution by sonication, and administered intraperitoneally at 0.1 ml/mouse twice a day at 9 a.m. and 6 p.m. for 4 consecutive days. For NAR administration, NAR (10 mg/0.1 ml PBS) or control PBS alone was intraperitoneally injected at the time of GNE-617 dosing. Unless otherwise specified, tumor tissues were isolated at 3 p.m. the day after GNE-617 dosing. A total of 15 and 25 NOG mice was used in experiments reported in Fig. 5a (also in Supplementary Fig. 4a, b) and 7b (also in Supplementary Fig. 9b), respectively; a total of 13, 34, 36, 3, 11, and 21 NOG mice was used in experiments reported in Figs. 5b–d, 6c, f, and g, respectively; a total of 30, 15, 14, 13, 14, 35, 20, 9, 17, 6, 11, 3, and 14 NOG mice was used in experiments reported in Supplementary Figs. 4c–e, 5a–k, 6–c, 7e, f, 8e, f, and 9c, respectively.

In NA administration experiments, mice fed a NFD (see below for details) for 3 days were then administered 4 mg NA/0.2 ml PBS by gavage at 9 a.m. on day 4. Blood samples were collected over time for up to 6 h after administration. For tracer experiments, [D4]-NA was administered by gavage and blood samples were collected 1 hour afterward. A total of 17 and 5 female NOG mice (8–12 weeks old) was used in experiments reported in Fig. 7c, d, respectively.

*Naprt*$^{flox}$ mice were generated by standard gene-targeting methods in mouse ES cells (on a C57/BL6 background). Obtained F$_1$ offspring (male) were backcrossed to C57/BL6JclN female mice, and further crossed with CAG-Cre mice (RikenBRC, Tsukuba, Japan) to produce mice carrying the *Naprt*$^-$ allele (in which *Naprt* exons 2–13 are deleted). *Naprt*$^{+/-}$ male and female mice were mated and resulting *Naprt*$^{WT}$ and *Naprt*$^{-/-}$ littermates were used in experiments. A total of 6 and 14 male mice (8–12 weeks old) was used in experiments reported in Fig. 7e, f, respectively. Both *Naprt*$^{flox}$ and *Naprt*$^-$ mouse strains were deposited with the CARD (Center for Animal Resources and Development) (Kumamoto, Japan).

Mice of the *Naprt*$^-$ strain were then crossed with a *Rag1*-KO line (The Jackson Laboratory, Bar Harbor, ME) to obtain double mutants. *Naprt*$^{+/-}$;*Rag1*$^{-/-}$ male and female mice were mated and resulting *Naprt*$^{WT}$;*Rag1*$^{-/-}$ and *Naprt*$^{+/-}$;*Rag1*$^{-/-}$ littermates were used in experiments. In transplantation experiments, only female mice (8-12 weeks old) were used. A total of eight mice was used in the experiment reported in Fig. 7g (also in Supplementary Fig. 9f).

## Mouse diets

The niacin-free diet (NFD) and a series of niacin-supplemented NFDs were produced and sterilized at Oriental Yeast Co., LTD. Synthetic diets were composed of 25% vitamin-free casein, 38% corn starch, 10% a-corn starch, 5% granulated sugar, 6% soybean oil, 8% cellulose, and a 6% mineral and 2% vitamin solution either containing or not containing niacin. Formulation of the vitamin mix was as follows (provided are amounts per 100 g vitamin mix): 0.15385 g vitamin A acetate (325 IU/ mg), 0.1 g vitamin D3 (100 IU/mg), 1 g vitamin E (50%), 0.52 g vitamin K3, 0.12 g vitamin B1, 0.4 g vitamin B2, 0.08 g vitamin B6, 0.05 g vitamin B12 (0.1%), 2 mg biotin, 20 mg folate, 0.5 g Calcium (+)-pantothenate, 0.6 g inositol, 0.5 g 4-aminobenzoic acid, 20 g choline chloride, 3 g vitamin C and 72.95415 g cellulose. The W (tryptophan)-free diet (WFD) was purchased from Research Diets Inc (New Brunswick, NJ). Niacin equivalent content of diets was determined by a standard bioassay using *Lactobacillus plantarum* ATCC-8014 at Japan Food Research Laboratories (Tokyo, Japan). The content of each niacin isoform contained in diets was determined by LC-MS analysis at the Foundation for Promotion of Material Science and Technology of Japan (Tokyo, Japan).

## Statistics and reproducibility

No statistical methods were used to predetermine sample size. No data were excluded from analyses. Experiments were not randomized, and investigators were not blinded to allocation during experiments and outcome assessment. MS analyses shown in Fig. 7a, qRT-PCR analyses shown in Figs. 4g and 6h, and the niacin test shown in Supplementary Fig. 9a were each performed once. All other experiments were performed at least twice. Student's *t* test (two-tailed) and a one-way ANOVA followed by a post hoc test were used when comparing two groups and multiple groups, respectively. A *P* value of <0.05 was considered significant.

## Reporting summary

Further information on research design is available in the Nature Portfolio Reporting Summary linked to this article.

## Data availability

Metabolome data used for generation of Fig. 2f and Supplementary Figs. 2a and 5a–k have been deposited in the "MetaboLights" database under accession codes MTBLS7251, MTBLS7252 and MTBLS7253 [https://www.ebi.ac.uk/metabolights/]. The CRISPR-KO and mRNA expression data used in this study (Figs. 2d, 3e, f and Supplementary Fig. 1h) are available in the "DepMap portal" [https://depmap.org/ portal/]. Source data are provided with this paper.

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

## Acknowledgements

We acknowledge Dr. Ikawa for providing Tg(CAG-Cre) mice. Thanks are also due to Ms. Y. Chiba for secretarial assistance, and K. Sato, M. Sato, and E. Mura-yama for help in the animal facility, H. Kato for help in histological analyses, Ms. Yui Tanuma for help in Figure illustration, and Dr. E. Lamar for English editing. This work was supported by JSPS KAKENHI grants (19H01036 and 21K19420 to N.T., 21K19748 to S.I., 21K09482 to H.Y., 21K16531 to N.K., 22K08989 to T.S., 19K09469 to Y.Y. and 20K17203 to M.M.), AMED P-Create under Grant number 23ama221122h0001 (to N.T.), Extramural Collaborative Research Grant of Cancer Research Institute, Kanazawa University   (to N.T.), the Takeda Foundation (to N.T.), the Uehara Memorial Foundation (to N.T.) and Princess Takamatsu Cancer Research Fund (to N.T.).

## Author contributions

N.T. conceived of and supervised the project. M.N. performed culture experiments in SCLC, NSCLC and organoid cells and analyzed data. M.O. and Y. Sakamoto performed overall cell culture and biochemical and animal studies. K. Kudo, K.H. and S.M. performed some animal experiments supervised by N.Y., H.Y. and H.S. K.Y. performed MS analyses of serum niacin and interpreted data under supervision of T.N. T. Soga and R.S. performed tumor metabolome analyses. Y. Sugiura performed metabolic analyses of mouse cells. M.M., Taku Sato, Y.Y., T.F., N.K., K.I.N., and A.H. analyzed the data. I.S. and S.I. performed tissue analyses. K. Kawasaki performed organoid experiments supervised by Toshiro Sato. Y.A. and S.A. performed PCa experiments and analyzed the data. S.I.T. synthesized, performed and interpreted TLM-118 NAMPTi experiments. M.O. and N.T. wrote the paper with input from all of the authors.

## Competing interests

The authors declare no competing interests.

## Additional information

[1]Division of Cancer Chemotherapy, Miyagi Cancer Center Research Institute, Natori, Japan. [2]Department of Biochemical Oncology, Tohoku University Graduate School of Medicine, Sendai, Japan. [3]Department of Obstetrics and Gynecology, Tohoku University Graduate School of Medicine, Sendai, Japan. [4]Department of Molecular and Medical Pharmacology, Faculty of Medicine, University of Toyama, Toyama, Japan. [5]Institute for Advanced Biosciences, Keio University, Tsuruoka, Japan. [6]Center for Cancer Immunotherapy and Immunobiology, Kyoto University Graduate School of Medicine, Kyoto, Japan. [7]Department of Pathology, Miyagi Cancer Center Hospital, Natori, Japan. [8]Department of Respiratory Medicine, Miyagi Cancer Center Hospital, Natori, Japan. [9]Department of Molecular and Cellular Biology, Medical Institute of Bioregulation, Kyusyu University, Fukuoka, Japan. [10]TMDU Advanced Research Institute, Tokyo Medical and Dental University, Tokyo, Japan. [11]Division of Molecular Genetics, Cancer and Stem Cell Research Program, Cancer Research Institute and WPI Nano Life Science Institute, Kanazawa University, Kanazawa, Japan. [12]Department of Organoid Medicine, Keio University School of Medicine, Tokyo, Japan. [13]Department of Urology, Miyagi Cancer Center Hospital, Natori, Japan. [14]Department of Urology, Kyoto University Graduate School of Medicine, Kyoto, Japan. [15]Department of Urology, Nagoya University Graduate School of Medicine, Nagoya, Japan. [16]Meikai University Research Institute of Odontology, Sakado, Japan. [17]University of Human Arts and Sciences, Saitama, Japan. [18]Department of Integrated Medicine and Biochemistry, Keio University School of Medicine, Tokyo, Japan. ✉e-mail: ntanuma@med.tohoku.ac.jp

