## [Peer Review File · Nature Communications]

REVIEWER COMMENTS

Reviewer #1 (Remarks to the Author); expert in metabolism:

The present interesting, well executed, and translationally relevant manuscript examines escape mechanisms from NAMPT inhibition in tumors, finding particular sensitivity of SCLCa, an important and sometimes understudied tumor type. In this tumor type, the main escape mechanism is uptake of a nicotinic acid related compound, which the authors interestingly identify as being NAR, not the more widely studied NR or NA. Overall, I am supportive for publication of the work without substantial changes, although some technical and methodological issues need to be addressed, specifically:

1. The PKM1 and M2 cell lines from the transgenic mice: How many independently derived cell lines were studied? From how many independent mice? And are these cell lines publicly available?
2. All experiments need to show individual data points, N, and define error bars. This was missing, among other places, for some of the key mouse experiments.

Reviewer #2 (Remarks to the Author); expert in neuroendocrine cancer:

In this study, Normura et al. build on previous work from their laboratory showing that small cell lung cancer (SCLC) uniquely expresses the PKM1 isoform of PKM. They first analyze metabolites selective for PKM1 over PKM2 in MEFs and lung epithelial cells expressing either PKM1 or PKM2 and find that high NAD was correlated with PKM1 expression. They then turned back to SCLC cell lines and found that SCLCs (relatively to NSCLCs) were more highly sensitive to siRNA depletion of NAMPT and also to a NAMPT inhibitor FK866. Additional studies identified differences in metabolites between SCLC and NSCLC. They then used an intestinal conditional neuroendocrine model to show that neuroendocrine differentiation correlated with high NAD and a NAMPT dependence. In vivo studies using NAMPT inhibitors with depletion of dietary niacin led to a marked reduction in tumor volumes across SCLC and NEPC models. They finally they do follow up studies to understand how Niacin depletion is necessary for the anti-tumor effects seen in the combination with niacin depletion and NAMPT inhibitors.

Overall these are novel findings that could be actionable therapeutically across different neuroendocrine tumors. They present a wealth of data with impressive in vivo efficacy data across models. I think this study has a lot of potential and would be of general interest to readers across the metabolism and neuroendocrine fields, but could be improved with the following experiments/suggestions:

1. The idea of NAMPT inhibition being a therapeutic vulnerability in neuroendocrine tumors is interesting and they provide some data on this with their in vivo data being particularly strong. Their in vitro data could be improved. They test multiple SCLC cell lines with only a single NAMPT inhibitor (FK866 and ? 1 concentration of drug which is not stated in the text) and only 1 siRNA against NAMPT1. It would be helpful to include 2 distinct NAMPT inhibitors in vitro and show full dose responses of the cell lines in panel 1c. Additionally, what do multiple (at least 2-3) different siRNAs and CRISPR/Cas9 sgRNAs look like in this cell line panel. It would also be helpful to present their data in a different way here (and throughout the manuscript) as most of their data is normalized and presented as heatmaps so it is unclear the quantitative effects and how many replicates were done. I want to really know how large the therapeutic window is between SCLC and NSCLC and how sensitive SCLC are relatively to what has been reported for other diseases such as AML?
2. The effects of QPRT as a driver of resistance to NAMPT inhibition in NSCLC is not convincing. Although statistically significant, the effects are minor in Fig. 3h and in the supplement. There are several instances in the paper where the data isn't that convincing and typically the authors interpretation is consistent with the results. I think the effects are overstated in the results section. How important do they think QPRT is as I don't think this explains the resistance to NAMPT in NSCLC? If it does not, then the interpretation of Fig. 3 changes.

3. Fig. 4-I like the DOX inducible neuroendocrine system but it is unclear how to interpret this model relative to neuroendocrine tumors. Also the data again is normalized and there are not benchmark controls (SCLC or NEPC cell lines) to understand really how neuroendocrine the model is. The authors should include data with benchmark controls for neuroendocrine high tumors in their studies characterizing the model. I think it is an interesting model if it truly is neuroendocrine but they need to present these data more clearly. I would say this is a general concern throughout the paper. Much of the data is normalized and therefore it is hard to interpret the findings. I would encourage the authors to move away from heatmap normalization and present their data in absolute numbers whenever possible with appropriate controls so we can quantitatively better interpret their data.

4. Fig. 5-The in vivo data is really strong overall with the combination of NFD + NAMPT inhibition being really encouraging. The WFD + NAMPT inhibitor was highly toxic but the reasons for this were unclear. Can the authors address the issues with this combination? It's surprising that the WFD + NAMPTi in Fig. 5a looks better than NAMPTi alone but the WFD + NAMPTi was off drug soon after the treatment started. Again, it would be helpful to analyze those tumors to understand this better? This should also be addressed in the discussion. Overall, I think the in vivo data is impressive but it's unclear whether this is a strategy that is feasible to achieve efficacy beyond mice. Are we underestimating the toxicity NAMPTi + NFD because of tolerability in mice that is species specific?

5. Data presentation: As mentioned in several points above, I think the paper could be improved by presenting the actual data as data points on either bar graphs or dose response curves rather than as heat maps. I think then the reader can then more transparently interpret the findings and the magnitude of the effects will be more clear.

RESPONSE TO REVIEWERS' COMMENTS

Answers to Reviewer #1

The present interesting, well executed, and translationally relevant manuscript examines escape mechanisms from NAMPT inhibition in tumors, finding particular sensitivity of SCLCa, an important and sometimes understudied tumor type. In this tumor type, the main escape mechanism is uptake of a nicotinic acid related compound, which the authors interestingly identify as being NAR, not the more widely studied NR or NA. Overall, I am supportive for publication of the work without substantial changes, although some technical and methodological issues need to be addressed, specifically:

We are honored to receive these comments recognizing our efforts. In this revision we now provide the requested missing information.

1. The PKM1 and M2 cell lines from the transgenic mice: How many independently derived cell lines were studied? From how many independent mice? And are these cell lines publicly available?

We revised description of MEFs and LE cell lines in the Methods section (page 15, lines 381-5) as follows:

MEFs and LE cells were established from Pkm-knock-in mice (PkmM1/M1 and PkmM2), transformed with oncogenic Kras, and cultured as described⁴. Mutant mouse strains were deposited with the RIKEN BioResource Research Center (Tsukuba, Japan). Three or 4 independent MEF and LE lines derived from 3 or 4 independent mice per genotype were used in this study.

We also note that those MEFs and LE cells are available upon request, as described in Data and material availability section of the Methods (page 20, line 523).

2. All experiments need to show individual data points, N, and define error bars. This was missing, among other places, for some of the key mouse experiments.

We carefully re-checked our manuscript and now provide information about N and error bars in Figure legends throughout the manuscript. Also, we provide all individual data used to produce all graphs as a Source Data file.

Answers to Reviewer #2

In this study, Normura et al... Overall these are novel findings that could be actionable therapeutically across different neuroendocrine tumors. They present a wealth of data with impressive in vivo efficacy data across models. I think this study has a lot of potential and would be of general interest to readers across the metabolism and neuroendocrine fields, but could be improved with the following experiments/suggestions:

We are grateful to receive these positive remarks on our study's significance as well as constructive comments. Prompted by this comment, we have now performed additional studies to validate our claim and made changes to improve transparency in this revision.

3. The idea of NAMPT inhibition being a therapeutic vulnerability in neuroendocrine tumors is interesting and they provide some data on this with their in vivo data being particularly strong. Their in vitro data could be improved.... It would be helpful to include 2 distinct NAMPT inhibitors in vitro and show full dose responses of the cell lines in panel 1c. Additionally, what do multiple (at least 2-3) different siRNAs and CRISPR/Cas9 sgRNAs look like in this cell line panel. It would also be helpful to present their data in a different way here ... it is unclear the quantitative effects and how many replicates were done. I want to really know how large the therapeutic window is between SCLC and NSCLC and how sensitive SCLC are relatively to what has been reported for other diseases such as AML?

3a) In response to these comments, we now provide data showing dose response curves of SCLC and NSCLC lines to FK866 (Supplementary Fig. 1e). We also added data using two additional NAMPTis (GNE-617 and TLM-118) (Fig. 2c). These analyses show that SCLC is more vulnerable not only to FK866 but also to other NAMPTi(s). We report these results on page 5, lines 113-4. We also note that sensitivity of lung cancer lines to the 3 NAMPT inhibitors is comparable, as shown in Figure A.

Figure A. Similarities of biological responses of lung cancer lines to FK866, GNE-617 and TLM-118. Results in Fig. 2c were further analyzed by constructing 2D plots.

3b) We newly analyzed effects of two additional NAMPT siRNAs (detailed information is provided in the Methods (page 16, lines 404-9)) and confirmed that all NAMPT siRNAs used here gave similar results. As seen in Supplementary Fig. 1b and c, NAMPT knockdown by the two additional siRNAs also decreased NAD levels in SCLC cells, as did the one originally used in Fig. 2b (referred to as siRNA #1 in Supplementary Fig. 1b and c).

3c) Analysis shown in Fig. 2d uses a public dataset of CRISPR/Cas9 loss-of-function proliferation screens (Project Achilles) collected in the DepMap. The library used in that project contains 4 sgRNA barcodes targeting the *NAMPT* gene, and screen results (namely, enrichment scores for each sgRNA) were then averaged, as reported in reference #10 (Meyers, RM. et al. Nat Genet 2017). We now state this explicitly in the Methods section (page 17, lines 432-3).

3d) We agree that comparing NECs and AML would be meaningful, and thus we now include analysis of some AML lines in FK866 experiments. As seen in Supplementary Fig. 1e and f, SCLC and AML lines overall showed similar FK866 dose response curves. We also analyzed the DepMap dataset and confirmed that the impact of *NAMPT*-KO on SCLC and AML lines is similar, and that effects were more robust than in the Pan-cancer group (Supplementary Fig. 1h). These results are stated on page 5, lines 114-5 and 117-9.

3e) We would like to clarify a more subtle point. In this revision, we incorporated GNE-617 and TLM-118 into *in vitro* experiments, but did not use FK866 in *in vivo* analyses. We made this decision because in mice, FK866 is rapidly cleared from the circulation after dosing—not because of its possible off-target effects, etc. Overall, we felt it would be difficult to maintain sufficient concentrations of FK866 *in vivo* to suppress NAMPT (Figure B).

Figure B. Serum concentrations of TLM-118 and FK866 in mice. Mice were dosed one time with 250 ug of either FK866 or TLM-118. We then obtained serum samples at the times indicated and analyzed them by MS spectrometry. The dashed line at 7.8 ng/ml is equivalent to 20 nM FK866.

Overall, this newly added information, experiments, and dataset analysis strengthen our conclusion that dependency on NAMPT is significantly high in NE lung and prostate cancers. Finally, based on the reviewer’s suggestion, we replaced heatmaps with Jitter plots in Supplementary Fig. 1d (also please see our reply to comment 7).

4. The effects of QPRT as a driver of resistance to NAMPT inhibition in NSCLC is not convincing. Although statistically significant, the effects are minor in Fig. 3h and in the supplement.

After the initial submission, we obtained strong evidence that QPRT indeed acts as a driver of NAMPTi resistance in SCLC cells. Specifically, forced QPRT expression rendered 87-5 and H209 lines resistant to FK866 in terms of NAD levels and proliferation/survival, as reported in new Figure 3i-k. We now report these results on page 6, lines 165-7 of the revision.

On the other hand, as the reviewer notes, the magnitude of QPRT knockdown effects in NSCLC cells was moderate, as shown in Fig. 3h. Currently, we suspect that other unknown factor(s) or mechanism(s) in NSCLC cells may mask effects of QPRT inactivation, leading to mild phenotypes (discussed on page 11, lines 298-304).

Given these mixed results, to report results more accurately, we changed the following sentences as indicated below:

(Results heading & Fig. 3 title): Absence of QPRT-dependent de novo NAD synthesis promotes NAMPTi-susceptibility in SCLC (page 6, line 141, page 25, lines 662-3)

(Results): These results indicate that de novo NAD synthesis, if activated, compensates for NAMPT inhibition in SCLC, and suggest that this activity partially accounts for NAMPTi-resistance in at least some NSCLC cells. (page 7, lines 167-9)

5. Fig. 4-I like the DOX inducible neuroendocrine system but it is unclear how to interpret this model relative to neuroendocrine tumors. Also the data again is normalized and there are not benchmark controls (SCLC or NEPC cell lines) to understand really how neuroendocrine the model is. The authors should include data with benchmark controls for neuroendocrine high tumors in their studies characterizing the model.

Based on these suggestions, in the revision, we now provide a comparison of NE marker expression between a Doxy-differentiated TR-6TF organoid and 3 NEC lines. As shown in Supplementary Fig 3, *SYP* mRNA levels were overall comparable, although *NCAM* mRNA levels appeared slightly lower in differentiated TR-6TF relative to the 3 NEC lines, although those differences were not statistically significant. We note, however, that expression of both *SYP* and *NCAM* mRNAs varies even in the 3 NEC lines.

We also note that we previously published more detailed phenotypic studies of this genetically-engineered organoid (reference #14, Kawasaki K et al, Cell 2020). For your information, we now enclose relevant pages of that paper as “reference material”; that material reports data relevant to morphology, Ki-67 levels, and global patterns of gene expression and chromatin status before and after differentiation of TR-6TF. We believe that it is generally accepted in the field that organoid

TR-6TF serves as a reliable system in which to analyze NE differentiation of tumor cells, and hope the information provided here helps evaluate this organoid.

6. Fig. 5-The in vivo data is really strong overall with the combination of NFD + NAMPT inhibition being really encouraging. The WFD + NAMPT inhibitor was highly toxic but the reasons for this were unclear... it would be helpful to analyze those tumors to understand this better? This should also be addressed in the discussion. Overall, I think the in vivo data is impressive but it's unclear whether this is a strategy that is feasible to achieve efficacy beyond mice. Are we underestimating the toxicity NAMPTi + NFD because of tolerability in mice that is species specific?

6a) The reviewer raises an important question. In response, we examined effects of a WFD on tumor NAD levels. As seen in Supplementary Fig 4d, combining GNE with a WFD synergistically and significantly decreased NAD levels in tumors. Mechanisms underlying this synergy remain unclear, but we believe that it could involve blood niacin levels: mice fed a WFD showed significantly lower levels of blood Nam and NAR (Supplementary Fig. 7f). Lowering of Nam by a WFD is consistent with the fact that Trp is a major dietary source of circulating Nam, yet mechanisms underlying NAR reduction remain unclear. Lower Nam/NAR would, in turn, impair NAD synthesis in tumor cells and synergize with NAMPTi, although these activities require further confirmation.

Lower blood Nam may also lead to WFD toxicity in host animals, since a NFD, which decreases NAR but not Nam, was tolerated by mice even when combined with NAMPTi treatment. However, other mechanism(s) may drive toxicity given that Trp is an essential amino acid. We believe, however, these issues are beyond the scope of study and should be addressed in future studies.

We state these findings in the Results (page 8, lines 203-5, and page 9, lines 241-2) and Discussion (page 12, lines 315-22) of the revised manuscript.

6b) As the reviewer notes, possible difference(s) between mouse and humans represent a potential limitation of this study, and clearly, this will be the next important question. Given this point, we added discussion of possible difference(s) between species in NAD/niacin metabolism (page 13, lines 334-6) to the revision. We are now conducting clinical studies to address these questions.

7.Data presentation: As mentioned in several points above, I think the paper could be improved by presenting the actual data as data points on either bar graphs or dose response curves rather than as heat maps. I think then the reader can then more transparently interpret the findings and

the magnitude of the effects will be more clear.

Thank you for these suggestions. We agree and have either modified data presentation or added figures to increase study transparency as follows;

- Two heatmaps in the previous version were replaced with Jitter plots in Fig. 4i and new Supplementary Fig. 1d.
- For heatmap presentations in Fig. 6a, 6e and Supplementary Fig. 8d, we provided individual data showing variation in repeated experiments as Jitter plots in Supplementary Fig. 7c/d, 8a and 8c.

REVIEWERS' COMMENTS

Reviewer #1 (Remarks to the Author):

Solid revision. Strong contribution to the literature.

Reviewer #2 (Remarks to the Author):

Thank you to the authors for thoroughly addressing all comments which in my opinion has substantially improved the manuscript. I have no further comments.

RESPONSE TO REVIEWERS' COMMENTS

Answer to Reviewer 1.

We thank you for recommending our paper for publication.

Answer to Reviewer 2.

We appreciate your review of our study. We believe that our revised manuscript is significantly improved thanks to your comments.